# Structure of the reduced microsporidian proteasome bound by PI31-like peptides in dormant spores

Nathan Jespersen[1,3], Kai Ehrenbolger [1,3], Rahel R. Winiger[1,3], Dennis Svedberg[1], Charles R. Vossbrinck[2] & Jonas Barandun [1] ✉

Proteasomes play an essential role in the life cycle of intracellular pathogens with extracellular stages by ensuring proteostasis in environments with limited resources. In microsporidia, divergent parasites with extraordinarily stream-lined genomes, the proteasome complexity and structure are unknown, which limits our understanding of how these unique pathogens adapt and compact essential eukaryotic complexes. We present cryo-electron microscopy structures of the microsporidian 20S and 26S proteasome isolated from dormant or germinated *Vairimorpha necatrix* spores. The discovery of PI31-like peptides, known to inhibit proteasome activity, bound simultaneously to all six active sites within the central cavity of the dormant spore proteasome, suggests reduced activity in the environmental stage. In contrast, the absence of the PI31-like peptides and the existence of 26S particles post-germination in the presence of ATP indicates that proteasomes are reactivated in nutrient-rich conditions. Structural and phylogenetic analyses reveal that microsporidian proteasomes have undergone extensive reductive evolution, lost at least two regulatory proteins, and compacted nearly every subunit. The highly derived structure of the microsporidian proteasome, and the minimized version of PI31 presented here, reinforce the feasibility of the development of specific inhibitors and provide insight into the unique evolution and biology of these medically and economically important pathogens.

Cellular proteostasis is the regulation of protein biogenesis, localization, and degradation in response to environmental stimuli, and involves the concerted efforts of several essential macromolecular assemblies. Proteasomes, proteolytic complexes expressed in all known eukaryotes, play a key role in this process by degrading undesirable and misfolded proteins. Eukaryotic 26S proteasomes are approximately 2.6 MDa in size and composed of ~33 different proteins. They can be divided into two sub-complexes: the 20S core particle (CP) and the 19S regulatory particle (RP). While the RP is responsible for substrate recognition, deubiquitination, unfolding, and translocation[1],

the CP houses the proteolytic active sites that hydrolyze targeted proteins. The barrel-shaped CP is arranged in four stacked hetero-heptameric rings, with subunits arranged in a C2-symmetric $\alpha_{1-7}\beta_{1-7}/\beta_{1-7}\alpha_{1-7}$ conformation[2]. The central cavity formed by these 28 subunits contains six active sites, in β1, β2, and β5 protomers, with caspase-like, trypsin-like, and chymotrypsin-like proteolytic activities, respectively[3]. In the presence of ATP, RPs bind to one or both ends of the CP to form the complete 26S or 30S proteasome[4]. RPs are sub-classified into base and lid complexes. The base contains a ring of six AAA-ATPases (Rpt1-6), three ubiquitin receptors (Rpn1, Rpn10, and Rpn13), and one

[1]Department of Molecular Biology, The Laboratory for Molecular Infection Medicine Sweden (MIMS), Umeå Centre for Microbial Research (UCMR), Science for Life Laboratory, Umeå University, 90187 Umeå, Sweden. [2]Department of Environmental Science, Connecticut Agricultural Experiment Station, New Haven, CT 06504, USA. [3]These authors contributed equally: Nathan Jespersen, Kai Ehrenbolger, Rahel R. Winiger. ✉e-mail: jonas.barandun@umu.se

structural protein (Rpn2)[5]. The lid is typically composed of a deubiquitinating enzyme (Rpn11) and eight structural proteins (Rpn3, 5, 6, 7, 8, 9, 12, and 15), which stabilize the 26S structure during dynamic substrate translocation.

Although proteasomes conserve nutrients by recycling amino acids, the process of unfolding and proteolyzing proteins requires an estimated 0.25-1.0 ATP per amino acid[6] and is often inhibited when nutrients are scarce. For example, when carbon is limiting, proteasomes reversibly accumulate in cytoplasmic puncta known as proteasome storage granules (PSGs)[7], which are thought to act as a reservoir from which proteasomes can be efficiently recovered once carbon is plentiful[8]. In other cases, proteasomes are inactivated through interactions with inhibitory proteins[9,10].

The 31-kDa proteasome inhibitor (PI31, also known as PSMF1 and Fub1) is a conserved eukaryotic protein that inhibits 20S proteasome activity and the ATP-dependent assembly of human 26S proteasomes in vitro but has no effect on pre-assembled 26S proteasomes[9–12]. Although these contrasting findings have provoked several conflicting hypotheses concerning the function and mechanism of PI31[9,13], a recent study succeeded in isolating and structurally characterizing the yeast PI31 ortholog in complex with the CP[10]. This work demonstrated that PI31 binds to the internal cavity of CPs and inhibits all six proteasome active sites simultaneously via a well-conserved, C-terminal proline-rich region. Interestingly, structural characterizations of the complex required the use of a mutated strain lacking the α3 subunit due to the low occupancy of PI31 in wild-type proteasomes. This mutation leads to the incorporation of two α4 subunits per alpha ring, and results in an aberrantly opened 20S gate conformation[10,14]. The increased interactions with open or incorrectly folded proteasomes suggest that PI31 may recognize and inhibit CPs with dysfunctional conformations in yeast[10]. It is unclear, however, whether PI31 plays a role outside of interactions with defective proteasomes, and if this function and mechanism are conserved across eukaryotes.

Microsporidia are highly divergent parasitic eukaryotes with host organisms described from most animal phyla[15]. They are notable for many characteristics, such as a lack of an innate motility complex[16], utility as a biological insecticide[17,18], and an unusual infection mechanism via injection through a tubular structure[19]. Despite their ecological[20], biomedical[21], and agricultural importance[17], they are perhaps best known for their widespread genome compaction, significant evolutionary divergence, and minimized macromolecular complexes[22–25]. In fact, the microsporidium *Encephalitozoon intestinalis* has the smallest known eukaryotic genome at only 2.3 Mbp[24], which is half the size of the *E. coli* genome (4.6 Mbp).

Microsporidia replicate within a host and have augmented their repertoire of import proteins to facilitate the utilization of host metabolites[26]. They are almost completely reliant on hosts for ATP generation, and are restricted to low levels of glycolytic activity for ATP production in the extracellular spore stage of their lifecycle[27]. It is therefore of paramount importance for microsporidian spores to conserve resources, and studies on ribosomes have identified several hibernation or dormancy factors that assist in the inhibition and eventual reactivation of ribosomes post-quiescence[23,28,29].

Here, we use cryo-electron microscopy (cryo-EM) to study the 20S proteasomes isolated from dormant microsporidian spores and identify a pair of PI31-like (PI31L) proteins symmetrically bound at high occupancy within the 20S central cavity. Like the yeast ortholog, PI31L occludes all six proteolytic active sites and adopts conformations incompatible with active digestion. In contrast to yeast PI31, PI31L is significantly truncated and follows an alternative path at several key interfaces, resulting in a minimized version of this important inhibitor. Interestingly, purification of proteasomes from spores after germination and under nutrient-rich conditions resulted in 20S structures lacking PI31L, suggesting PI31L acts as a hibernation factor to inhibit

background degradation in metabolically inactive microsporidian spores, but is removed when nutrients are plentiful.

The existence of a small number of singly- and doubly-capped proteasomes in the germinated sample, which were absent in dormant spore samples, allowed us to generate an architectural model of the 26S proteasome. Our data are consistent with a highly modified 26S structure containing a notably truncated Rpn3 and missing Rpn12 and Rpn13 at the genomic level. A comparison of yeast and microsporidian proteasomes demonstrates that, while microsporidia have retained the expanded repertoire of 20S subunits characteristic of eukaryotic proteasomes, they have eliminated many of the insertions which distinguish individual α and β subunits. These deletions and truncations have resulted in small-scale structural rearrangements throughout the proteasome, most importantly at the CP gate and in several regions interfacing with the RP. Although active site residues responsible for proteolysis are well conserved, the specificity pocket of the *V. necatrix* β5 subunit adopts an open conformation associated with selective inhibition by several α', β' epoxyketone compounds, highlighting the utility of these compounds as possible therapeutics. Finally, a phylogenetic analysis using proteasomal subunits reinforces previous taxonomic groupings and emphasizes the structural streamlining in microsporidian macromolecular complexes.

## Results

### Isolation and structural characterization of microsporidian proteasomes

The study of microsporidian macromolecular complexes is complicated by a lack of genetic tools to tag or modify proteins of interest and an inability to culture microsporidia in a non-cellular growth medium, leading to limitations in starting material and requiring the use of traditional enrichment procedures. To study proteasomes derived from the dormant state, we ruptured spores via bead-beating, in the absence of ATP, and performed size exclusion chromatography on the spore lysate. Fractions enriched in proteasomes were analyzed via mass-spectrometry (Supplementary Data 1) and cryo-EM, resulting in particles consistent with both top and side views of the 20S proteasome. No 26S proteasomes were identified, despite previous data indicating that RP proteins are present in spores[30]. This suggests that the CP and RP are not associated in dormant spores lacking ATP. Using 52,679 particles for 3D refinement, we obtained a 2.8 Å map of the 20S CP (Fig. 1, Supplementary Figs. 1 and 2, Supplementary Table 1). Although previous work has noted difficulties in matching proteasomal orthologs in microsporidia due to low sequence identities[31], the resolution of our data allowed us to unambiguously identify all CP subunits (Supplementary Table 2). Similar to typical eukaryotes, microsporidian proteasomes are composed of 28 proteins in a C2-symmetric $\alpha_{1–7}\beta_{1–7}/\beta_{1–7}\alpha_{1–7}$ arrangement. Most regions involved in essential functions, such as the catalytically active residues and substrate recognition pockets, are structurally well-conserved. On the other hand, many short loops and disordered tails are absent. Additionally, within the central cavity of the CP barrel, we identified symmetric density that spans multiple β subunits and blocks the active sites of β1, β2, and β5 (Fig. 1c), suggesting this density does not represent an actively digested substrate.

### Proteasomes from dormant spores are bound by PI31-like peptides

Proteasome CPs contain three connected cavities through which unfolded proteins pass: two antechambers created by the α-rings, and a catalytic chamber housing the active sites of the β-rings (Fig. 1b, c, 2). These enzymatic active sites are N-terminal nucleophilic hydrolases that use a deprotonated threonine hydroxyl group to cleave peptide bonds. Activation of the threonine residues in subunits β1, β2, and β5 requires the cleavage of propeptides after highly conserved glycine residues[32]. Substrate preference varies between the three protease

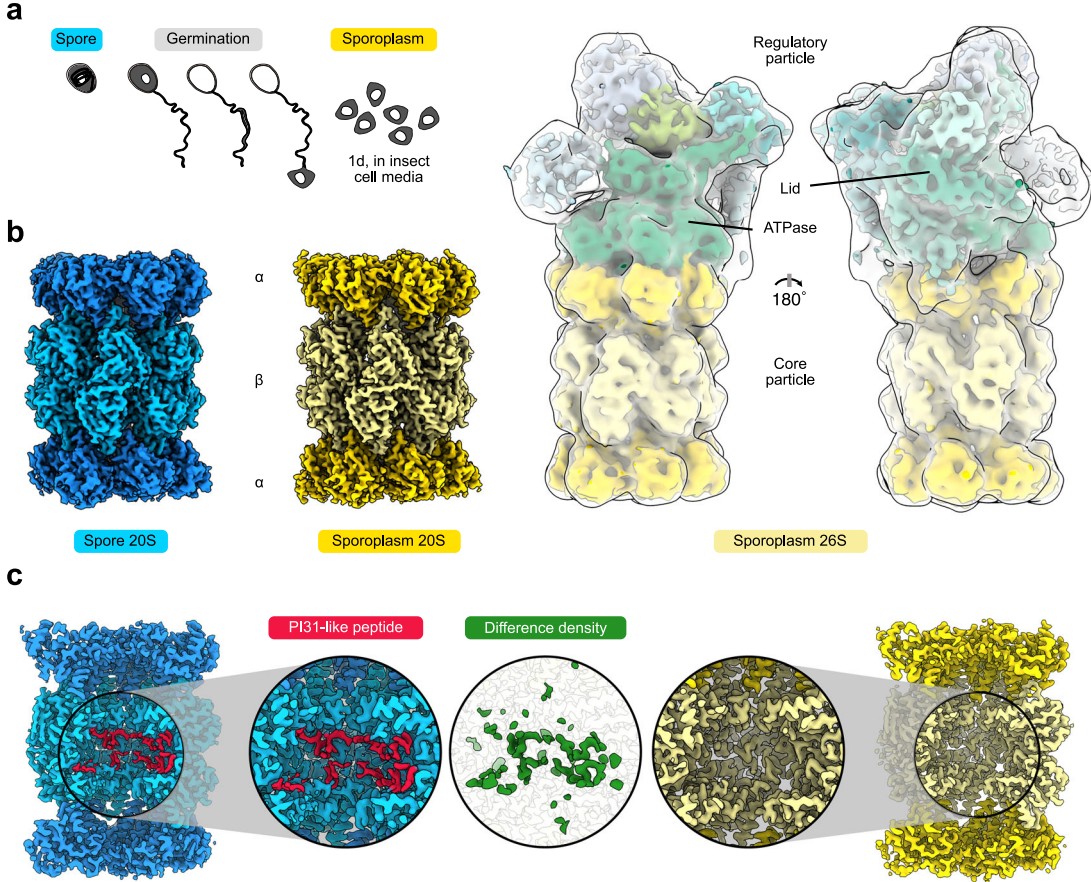

**Fig. 1 | The cryo-EM structure of the microsporidian proteasome, isolated from spores or sporoplasms. a** Schematic overview of the germination process of a microsporidian spore proceeding from the environmental stage to the release of the sporoplasm through the polar tube. The stages from which proteasomes were isolated are labeled in blue (spore stage) and yellow (sporoplasm stage). **b** Cryo-EM density of spore-isolated (blue) and sporoplasm-isolated (yellow) proteasome reconstructions. The 8.3 Å 26S cryo-EM density is shown with RP subunits colored in shades of green, enclosed in a transparent white 16-Å lowpass-filtered map. **c** Slab-view of the full proteasome and zoom-in sections of the central proteolytic cavities. The peptide-like density (red) corresponding to PI31L, is present in the spore (left) but absent in the sporoplasm isolated proteasome (right). The central circular inlet shows the difference density (green) between 4-Å lowpass-filtered cryo-EM maps from spores and sporoplasm proteasomes.

subunits and is determined by residues in each S1 specificity pocket[2]. Catalytic residues are identical in microsporidia and other eukaryotes, indicating that microsporidian proteasomes utilize a cleavage strategy similar to other eukaryotes. Additionally, the S1 specificity pockets are largely conserved, demonstrating sequence specificities of *V. necatrix* proteasomes mirror those of other eukaryotes.

The cryo-EM map of our spore-isolated proteasome displays pronounced additional density in the central chamber (Figs. 1c, 2a). Multiple peptide-like densities are present, with the most prominent and continuous density found in the catalytic chamber. Each antechamber contains two poorly-resolved fragments, while each β-ring houses three well-resolved sections: two shorter sections of 11 and 14 residues and one more extended section of 34 residues (Fig. 2a). The related peptide fragments are nearly indistinguishable in volumes refined with and without C2 symmetry (Supplementary Fig. 1), indicating that binding occurs symmetrically and specifically. Interestingly, the observed peptide-like densities share some similarities with a recently published yeast 20S structure containing the proteasome inhibitor PI31[10]. Using the yeast version of PI31 as a starting point, we utilized a combination of motif searches and structural homology searches to identify divergent PI31-like proteins in microsporidians (Supplementary Fig. 3, Supplementary Data 2). The identified PI31L of *V. necatrix*, a small protein of 147 amino acids, fits in the available density. Our model includes residues 76-89 and 96-106, corresponding to the two shorter densities, and residues 113-147, corresponding to the

well-resolved continuous section (Fig. 2a, b and Supplementary Fig. 3). The N-terminal half (1-75) and two short linker regions (90-95, 107-112) are not resolved. Our assignment of PI31L was further supported by a mass spectrometry analysis, which demonstrated high coverage for the identified protein in 20S proteasome samples (Supplementary Data 1).

Two PI31L molecules bind with their C-terminal half (residues 76 to 147) symmetrically in the central cavity and occlude all 3 active sites in each ring (Fig. 2, Supplementary Fig. 3b). The resolved C-terminal portion of the protein starts on β5 and β6 with the two shorter fragments (76-89, 96-107), connected through a disordered linker of 6 amino acids (Fig. 2a, c). A short flexible linker connects these shorter fragments to the continuous density corresponding to amino acids 113-147, located on the opposite β-ring. There, PI31L blocks the active sites of β2 and β1 and buries its C-terminal end in binding sites on β7.

Residues of PI31L occupying the active sites are denoted as P4-P3-P2-P1-P1'-P2'-P3'-P4', where digestion occurs between P1 and P1'. Intriguingly, these residues adopt unique folds at each site to evade degradation and show some distinct differences from yeast PI31 (Fig. 2). In the β5 pocket, PI31L bulges out and loops back in on itself, resulting in Phe101, Ile100, Ala99, Asp84, and Leu85 representing P3, P2, P1, P1', and P2', respectively (Fig. 2d, e). This unusual arrangement causes the proteolytically active site to be occupied by noncontiguous amino acids and serves to inhibit proteolytic function. Although the general structure of PI31 is conserved in this region between yeast and

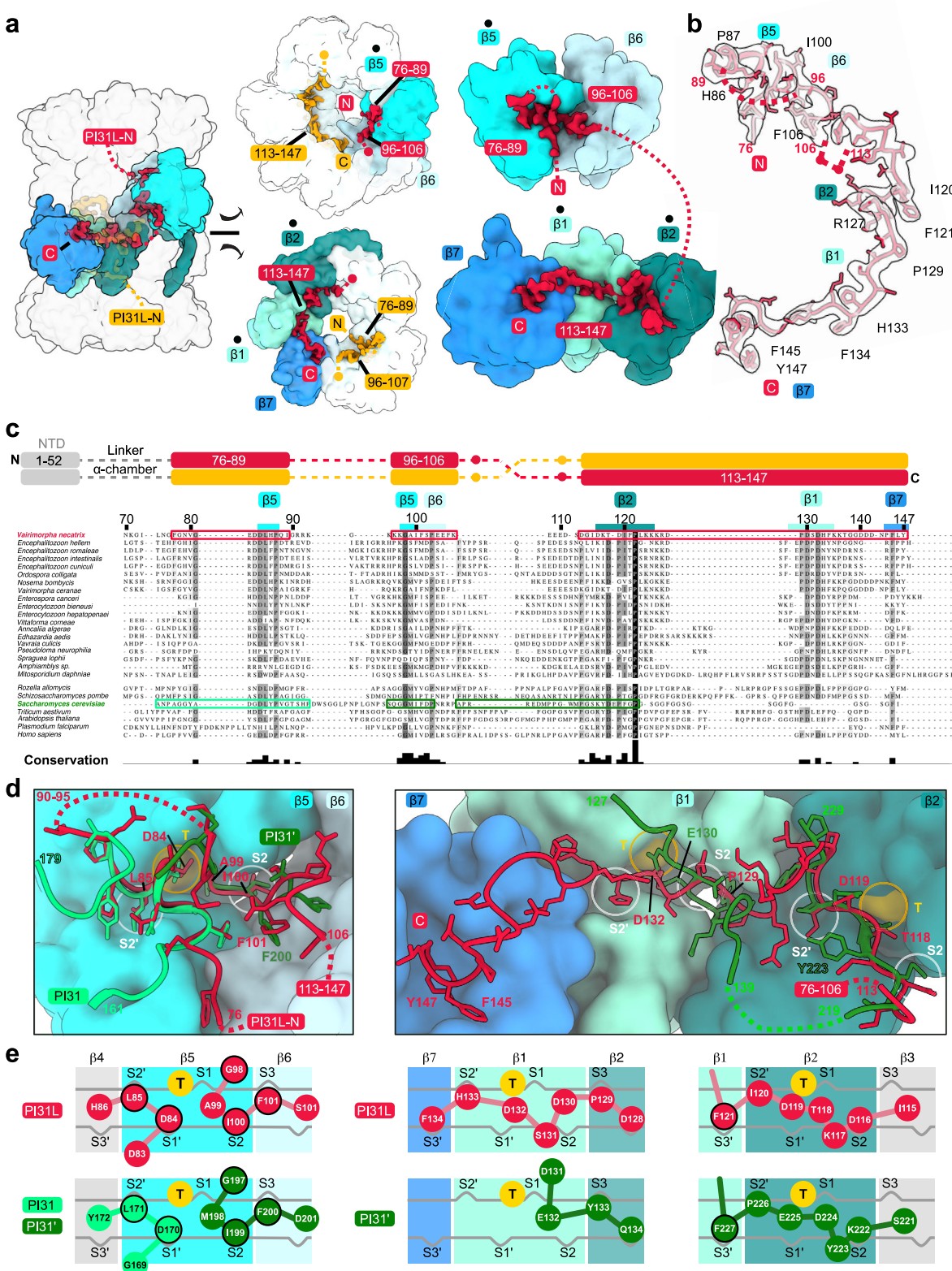

*V. necatrix*, previous work suggested that the noncontiguous residues are derived from two different PI31 protomers which cross the central cavity (Fig. 2d)[10]. In the PI31L structure, a poorly-resolved bridge is visible between residues Ile89 and Lys96 of the same protomer. While these six residues are sufficient to span the 13 Å distance between Ile89 and Lys96 on one β5 protomer, they are insufficient to cross the 31 Å gap between Ile89 and Lys96 on opposite faces of the barrel (Fig. 2a, d,

Supplementary Fig. 3c). Taken together, our data support an alternative structural arrangement for the significantly truncated PI31L in proximity of the β5 subunit than what has been previously described[10] for the more extended yeast version (Fig. 2c).

At the trypsin-like β2 site, which typically cleaves after basic residues, the S1 and S1' sites are partially occupied by Thr118 and Asp119, respectively. The lack of a basic residue, in combination with

**Fig. 2 | Spore proteasome active sites are blocked by PI31-like peptides.**
**a** Overview of PI31L (red solid and orange transparent cryo-EM densities) in the context of the full proteasome (left) and views of the two β-rings, next to selected PI31L-bound β-subunits. The selected proteasome subunits are shown with surface representations, colored in shades of blue and superimposed with the cryo-EM densities of PI31L (red, orange). Subunits marked by a black dot highlight those responsible for proteolysis. **b** Cryo-EM density and model fit of one PI31L peptide, with selected residues labeled. **c** Domain architectures of the two PI31L peptides are shown above a sequence alignment of the C-terminal domain of the identified microsporidian PI31L peptides with a selection of eukaryotic PI31. Modeled sections are indicated with colored bars or red squared sequence stretches, while disordered loops are depicted with dotted lines. (**a**, **b**, **c**) The location where PI31L traverses into the opposite ring is indicated with dotted lines and an orange/red circle. **d** Zoom-in sections of active sites of the proteolytically active β-subunits. The β-subunits are shown as surfaces colored as in (**a**), with the active-site threonine highlighted in yellow and PI31L as red cartoon with selected residues as sticks. The yeast PI31 structure (PDB-7TEO[10] [https://www.rcsb.org/structure/7TEO]) is superimposed and shown in shades of green. **e** A schematic representation of proteolytic active sites and specificity sites, with active site threonines shown in yellow and PI31 orientations shown in red (*V. necatrix*) or green (*S. cerevisiae*). Residues conserved in yeast and *V. necatrix* are circled in black.

interactions between PI31L and surrounding β2 residues, results in the P1 residue being ineffectively slotted into the active site (Fig. 2d, e). The carbonyl carbon that would be targeted during the nucleophilic attack is almost 6 Å from the hydroxyl group of the active site threonine (Fig. 2d, Supplementary Fig. 3d). This structural arrangement appears to be well conserved between yeast PI31 and microsporidian PI31L. In contrast, PI31L binding at the caspase-like β1 site varies significantly. Yeast PI31 passes at a different angle from C- to N-terminus through the active site, with the P1-site carbonyl 6.9 Å from the active site threonine (Fig. 2d, e, Supplementary Fig. 3e). In contrast, *V. necatrix* PI31L is inverted and passes horizontally through the active site of β1 in an N- to C-terminal orientation (Fig. 2d, e). The P1 site carbonyl group is closer to the active site (~4.9 Å). However, PI31L appears to evade digestion via structural constraints caused by Pro129 in the P4 site, which serve to push the P1 site Asp132 out of the active site and into a region where the angle is suboptimal for nucleophilic attack by the active site threonine. The density in the β1 active site region is the most well-resolved part of the PI31L structure, suggesting structural rigidity is important for avoiding digestion. The terminal residues of PI31 differ significantly between yeast and other eukaryotes (Fig. 2c), and in *V. necatrix*, they are buried in pockets on the β7 subunit (Fig. 2).

## PI31L is absent in proteasomes purified under nutrient-rich conditions

To determine if PI31L acts as a hibernation factor and is removed in nutrient-rich conditions, we activated 10 mg of microsporidian spores through germination using alkaline priming[33]. We then incubated the germinated spores for 1 day in an ATP-rich insect cell medium to more closely mimic an intracellular environment, lysed the fragile sporoplasms via sonication, and purified proteasomes in the presence of ATP using differential centrifugation. Both 20S and 26S particles were clearly visible in these samples via cryo-EM (Supplementary Fig. 1), which allowed us to obtain a volume of the CP at a resolution of 3.2 Å (17,942 particles), as well as a low-resolution volume (~8.3 Å; 6,442 particles) of the 26S proteasome (Fig. 1b). The overall structure of the spore and sporoplasm isolated 20S proteasomes are essentially identical. Both proteasome pores are closed, and the catalytic chamber is not accessible. In contrast to the overall structural similarities, the sporoplasm core particle entirely lacks the PI31L densities found in the spore 20S (Fig. 1c). Although the importance of ATP for CP-RP association is well described[4], PI31 is known to inhibit the ATP-dependent formation of 26S proteasomes[9], suggesting that a combination of factors is required for the removal of PI31L.

## Conservation and variation in 20S proteasome active sites

Residues in β subunits adjacent to enzymatic subunits contribute to the structure of S1 pockets and help form S3/S4 specificity sites, which are often targets for the development of selective proteasome inhibitors[2,34]. This region is more divergent in *V. necatrix* (Fig. 3), with large changes to the charge distribution around the β2 (trypsin) site. The negatively charged S3/S4 specificity site residues, which stabilize the positively charged substrates, are largely replaced with neutral or positive residues, providing a potential target for the development of selective therapeutics. Notably, the β5 S1 site methionine (Met76) is retracted, potentially due to van der Waals interactions with adjacent Tyr and His residues in β6[35], creating a larger binding pocket (Supplementary Fig. 4). Previous work has established that the relative orientation of this methionine (Met45 in yeast) is important for the binding of several inhibitors[35,36]. In fact, several selective proteasome inhibitors like PR-957 have a 15- to 255-fold higher affinity for mammalian immunoproteasomes than for constitutive proteasomes[35,36]. The methionine of immunoproteasomes is in an 'open' conformation, allowing the efficient insertion of a bulky sidechain into the S1 site. In constitutive proteasomes, on the other hand, Met45 is in a 'closed' conformation that sterically inhibits PR-957 interactions[35]. The retracted methionine structure visible in *V. necatrix* proteasomes is consistent with this open conformation (Supplementary Fig. 4). PR-957 and related compounds may therefore represent a reservoir of known inhibitors useful for the selective targeting of microsporidian proteasomes. In contrast, several other classes of known inhibitors are unlikely to bind in microsporidia. Thiol-reactive maleimides, for example, were designed to covalently bind to a well-conserved Cys118 in β3[34,37]. The mutation of this residue to an alanine in *V. necatrix* (Supplementary Data 2) may therefore result in a low affinity for this class of inhibitors.

## Microsporidia evolved a different gate access mechanism

The substrate access to the proteolytic chamber is tightly controlled through gated pores in the α-rings[38]. The association of accessory proteins harboring C-terminal hydrophobic-tyrosine-X (HbYX) motifs, such as the RP, induces a conformational change in the N-terminal tails in the α-ring subunits[39]. A conserved YDR motif (Fig. 4a) is involved in the formation of the open-gate conformation through specific interactions between the tyrosine and a proline, present in all subunits[38].

Microsporidia have significantly changed the N-terminal tails of the α subunits (Fig. 4a, b) and evolved an alternative mechanism to restrict access. Overall, the tails are shorter than in yeast orthologs and adopt a different closed-gate conformation. In yeast, the N-terminal tails of multiple subunits (α2 to α5) extend into the pore and block the access. The tails of α2 and α4 descend into the pore while α3 and α5 sit on top (Fig. 4c). In *V. necatrix*, the closed gate is formed predominantly by the α2 and α5 tails. The tail of α2 takes over the function and space of yeast α2 and α3 tails, and a short helical segment in α5 serves as a plug. The pore of this gate is less rigorously closed, resulting in a small hole visible between α2 and α4 (Fig. 4). The conserved YDR motif and the following proline, required to stabilize the open-gate conformation, have also been lost in all α-subunits except for the Tyrosine in α6 (Fig. 4a). Interestingly, deletions of the first nine amino acids of α3 in yeast are associated with permeable 20S proteasome gates, which allow for background proteolytic activity[38]. The truncation of seven amino acids at the N-terminus of α3 in *V. necatrix* may therefore contribute to the less stringently closed proteasome gate, which suggests an intriguing potential role for PI31L as an inhibitor of background proteasome degradation in dormant spores.

The C-terminal HbYX motifs of the ATPase subunits Rpt2, Rpt3, and Rpt5 are only partially retained in microsporidia (Fig. 4d,

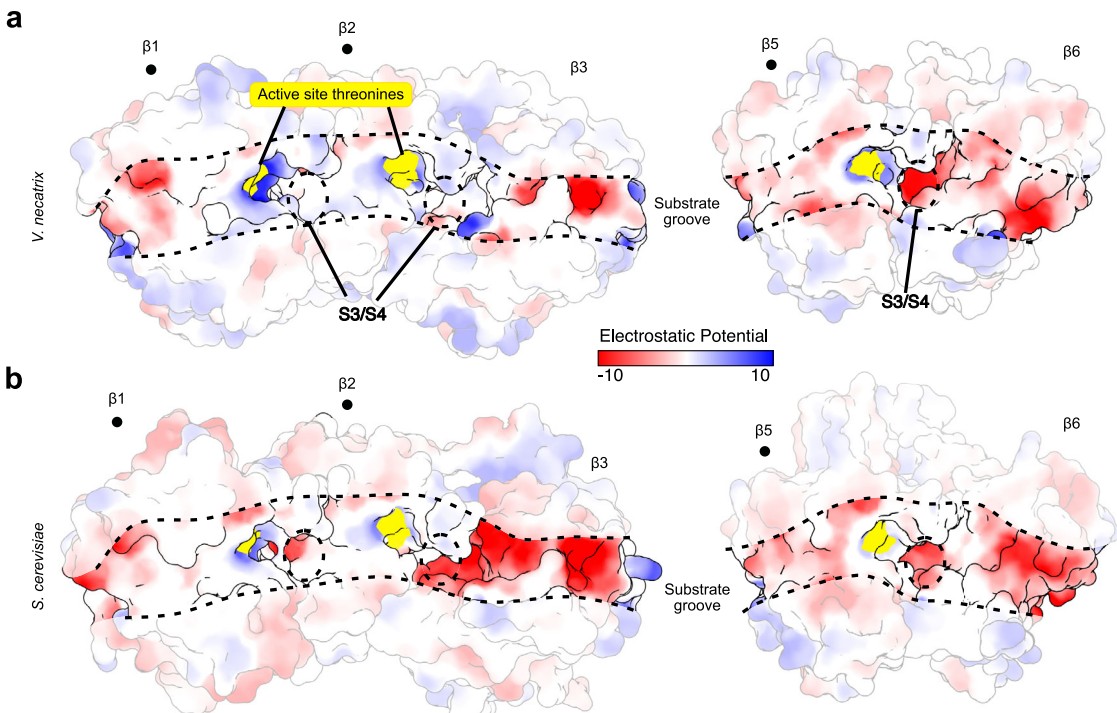

**Fig. 3 | Electrostatic potential for substrate binding grooves.** Surface representation of subunits important for substrate specificity in the proteolytic active sites of *V. necatrix* (**a**) or *S. cerevisiae* (**b**) (PDB-5CZ4[32] [https://www.rcsb.org/structure/5CZ4]), colored based Coulombic electrostatic potential in kcal/(mol·e). Active site residues are colored in yellow. Subunits harboring active sites are denoted with a •. The S3/S4 specificity regions are circled for emphasis.

Supplementary Data 2), suggesting analogous, but non-identical, binding interactions between the RP and CP are present. Recent work has demonstrated the importance of other Rpt C-termini in the binding and gate-opening of the 26S proteasome[40]. While N- and C-terminal deletions are present in most microsporidian proteasomal orthologs, the Rpt subunits terminate at the same residue as their yeast orthologs, with the exception of a two-residue truncation in Rpt3 (Fig. 4d, Supplementary Data 2). These data support a conserved role for the C-termini of Rpt proteins in binding and gate-opening of microsporidian proteasomes.

**The architecture of the microsporidian 26S proteasome**
Proteasomes isolated from germinated sporoplasms in nutrient-rich conditions were capped by RPs in about 12% of all particles, with a small portion of proteasomes (1.5%) capped on both ends. Despite limitations in starting material and the small total population of capped proteasomes, we were able to generate a low resolution (8.3 Å) map of the 26S proteasome from *V. necatrix*. To structurally characterize the complex, we identified microsporidian orthologs, predicted tertiary structures using AlphaFold[41,42], and superimposed models onto the closest homologous structure from yeast. Models were then manually adjusted to best fit the volume. *Vairimorpha necatrix* proteasomes display a similar general structure to 26S proteasomes from other eukaryotes, with the barrel-like CP capped by a ring of six ATPases and a "grasping-hand"-like lid complex (Fig. 5a)[43]. Interestingly, there is a conspicuous lack of density for two RP subunits: Rpn12 and Rpn13 (Fig. 5b). Although Rpn13 is often poorly-resolved[43], our findings are corroborated by previous work that was unable to identify microsporidian orthologs for these proteins[31]. Additionally, no density is identifiable for Rpn15, nor is it identifiable via BLAST[44], suggesting it may also be absent in microsporidia. However, due to the small size and poor sequence conservation for Rpn15, as well as the low resolution in this area of our map, we cannot exclude the possibility that a divergent version of Rpn15 is present in microsporidian proteasomes. In contrast, clear density for Rpn3 is visible,

and a highly truncated version of Rpn3 is identifiable using BLAST[44] (Fig. 6, Supplementary Data 2), despite prior speculation that it is absent in microsporidia[31].

The truncation of Rpn3 and absence of Rpn12 alters the overall lid structure, and the "grasping-hand" in microsporidian proteasomes has a smaller thumb and a much more open purlicue (Fig. 5). It is unclear how this more open structure affects enzymatic activity or proteasome stability; however, there is a surprisingly minor effect on proteasome structure in other regions. For example, in most eukaryotes the C-terminal tails of the eight structural lid proteins (Rpn3/5/6/7/8/9/11/12) all congregate in a central and essential helical bundle[45]. The final helix integrated, that of Rpn12, induces widespread structural remodeling of the lid and elicits the formation of the complete RP complex[46]. The absence of Rpn12 in *V. necatrix* precludes the integration of its helix. Nonetheless, the proteasome adopts a conformational state akin to the final remodeled structure in yeast and displays a clear central density that likely represents the helical bundle.

Microsporidia have also reduced subunits responsible for the recognition and deubiquitination of substrates. Rpn13, one of three ubiquitin receptors in the RP, is absent in our structure and is not identifiable in any microsporidian species (Fig. 5, Supplementary Data 2)[31]. Importantly, deletions of Rpn13 in yeast are nonlethal[47]. Rpn13 mutations in yeast are instead associated with a two-fold decrease in binding affinity to ubiquitin-conjugated substrates[48,49], suggesting this deletion reduces the ubiquitin binding potential of microsporidian proteasomes. Additionally, a comparison of Rpt5 sequences in yeast and *V. necatrix* reveals that the well-conserved 'N-loop' has been removed (Residues 88-114 in yeast; Supplementary Data 2). This loop directly interacts with ubiquitin-bound Rpn11 in human proteasomes, stabilizing the interaction and optimizing the positioning of the isopeptide bond for efficient deubiquitylation[50]. These examples demonstrate that microsporidia have streamlined proteasome structure, and removed segments associated with important, but nonessential functions.

A comparison of the quaternary arrangement of *V. necatrix* subunits to six known yeast conformational states indicates our structure is most

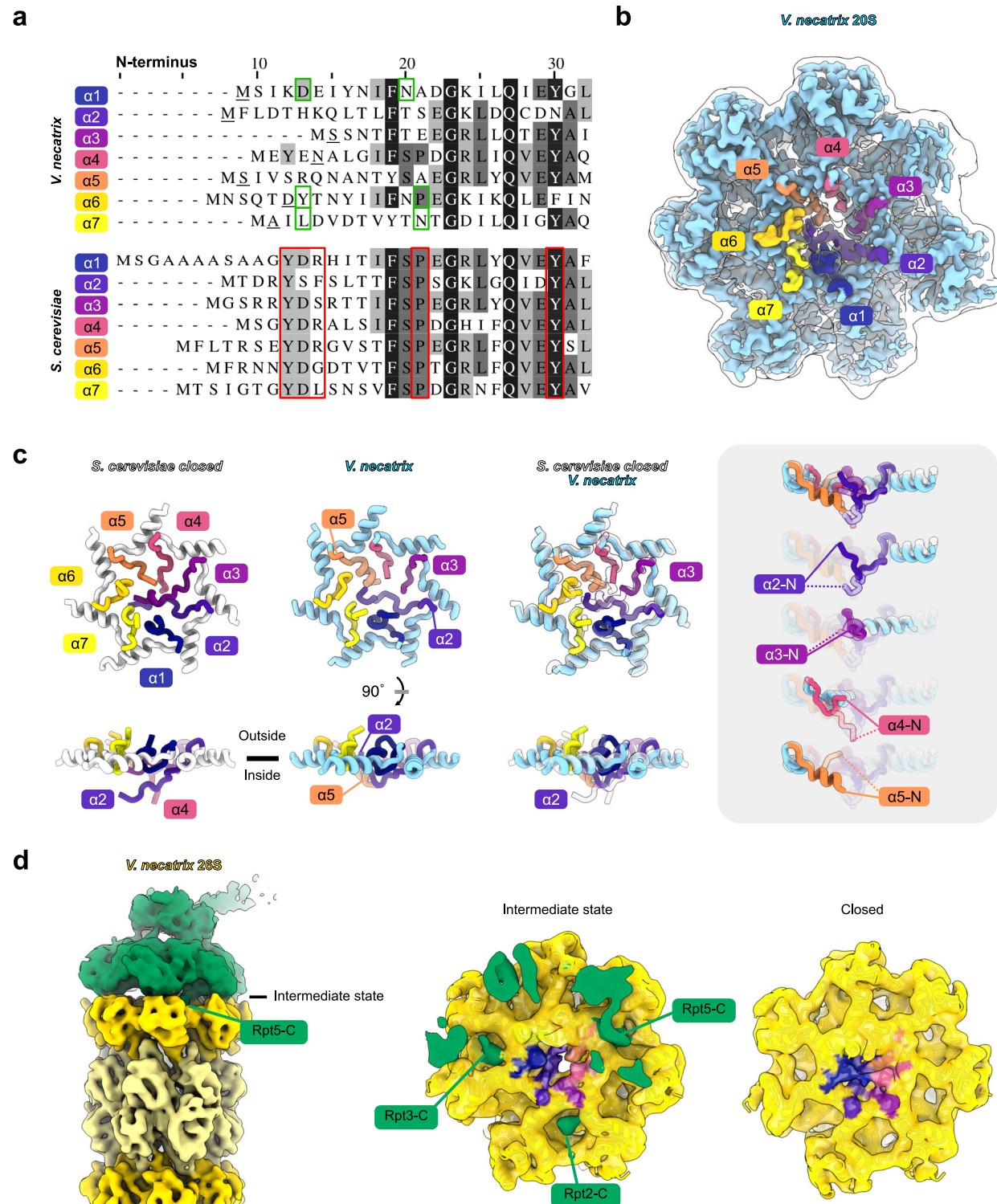

**Fig. 4 | Microsporidia lack the conserved YDR motif and differ in gate lining α-subunit tail structure. a** Alignment of the *V. necatrix* (top) and *S. cerevisiae* (bottom) N-terminal *α*-ring subunit tail amino acid sequences. Critical residues for the open-gate formation in eukaryotes are lined with red borders, the first resolved residues in the *V. necatrix* cryo-EM density are underlined in black, and the residues involved in an open-gate like conformation in the 20S structure are boxed in green. Sequence conservation is indicated with shaded residues from white for low conservation to black for highly conserved positions. **b** The 20S cryo-EM map with the N-terminal tails, colored and labeled as in **a**, is depicted in solid and superimposed with a transparent white lowpass-filtered map. **c** Structural models of the N-terminal tails of the α subunits in *S. cerevisiae* (PDB-5CZ4[32] [https://www.rcsb.org/structure/5CZ4]) and *V. necatrix*[32], shown top down and from the side. The right side depicts superpositions of isolated chains in *S. cerevisiae* (transparent) and *V. necatrix* (solid) to emphasize differences in individual N-terminal tails. **d** The ATPase module (green) and the core (shades of yellow) of the 26S cryo-EM density are shown in isolation next to a slice through the RP-CP interaction interface (middle) and a view from the bottom (right). The tail regions are colored as in (a-c).

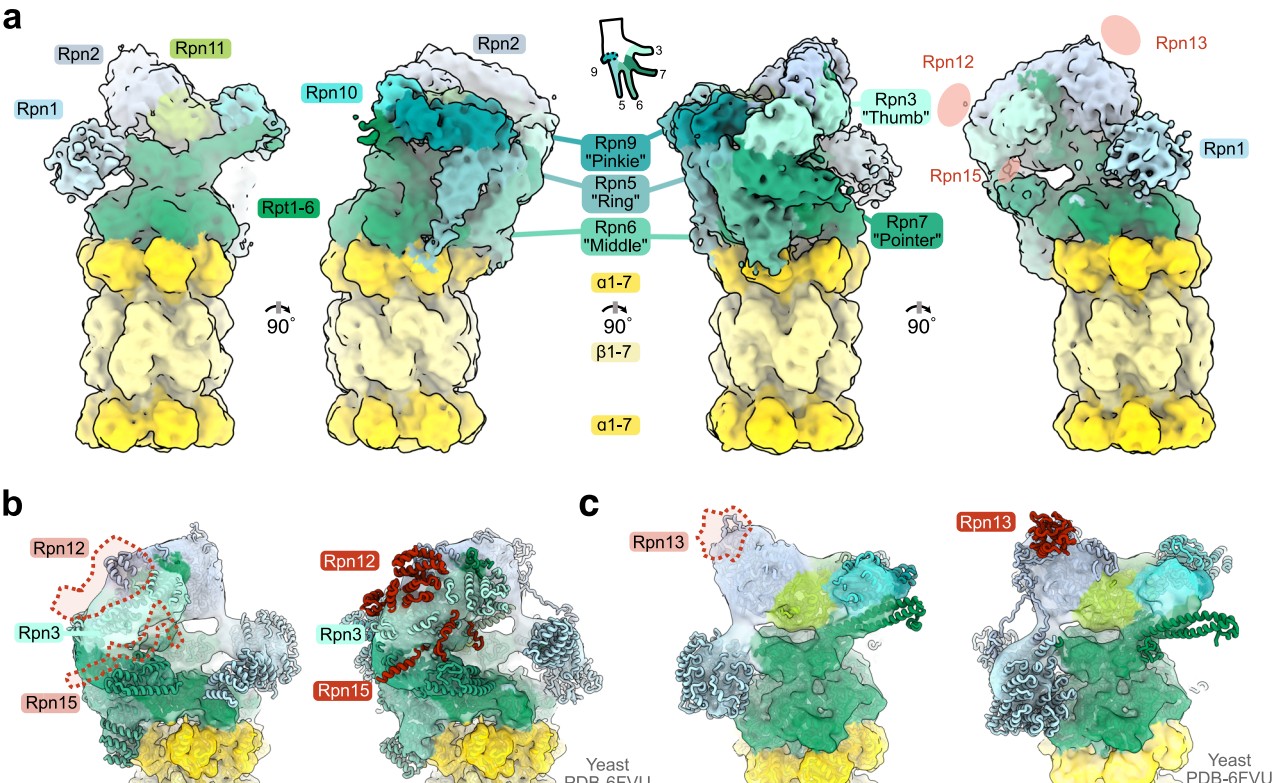

**Fig. 5 | The architecture of the microsporidian 26S proteasome and missing regulatory subunits. a** Four 90°-rotations of the sporoplasm-isolated 26S cryo-EM density are shown with the regulatory particle components colored in shades of green and the core particle indicated in shades of yellow. Subunits in the "grasping-hand" are indicated in the hand schematic and labeled on the middle two structures. The location of eukaryotic subunits not observed in the microsporidian proteasome density is indicated in the right-most view. **b, c** Two different views of the microsporidian regulatory particle cryo-EM density are shown with docked AlphaFold[41] models (left) and the *S. cerevisiae* structure (PDB-6FVU[40] [https://www.rcsb.org/structure/6FVU]). Missing elements are indicated with dashed lines and colored red in the yeast structure.

consistent with the s2 state[40]. Proteasomes in the s2 state are considered primed for degradation, as the CP, ATPase ring, and Rpn11 are appropriately aligned for substrate deubiquitination and eventual degradation. The CP gate is largely closed in the yeast s2 conformation, leaving a gap of only ~5 Å that hinders access to the catalytic chamber[40,51]. The opened s4 gate, in comparison, is ~17 Å wide. Although the majority of our structure is reminiscent of the yeast s2 state, the gate occupies an intermediate conformation between opened and closed, with a gap diameter of ~11 Å. It should be noted, however, that the conformational landscape of the 26S proteasome is diverse, with CP gates often sampling both closed and opened conformations[52]. The limited particle numbers in our sample hindered our ability to subclassify 26S proteasomes further, and the intermediate state identified here may therefore represent an average of several conformations. The C-terminal HbYX motifs of Rpt2, Rpt3, and Rpt5 are docked within the α-ring of the CP, as is typical[39], despite the small C-terminal truncation in Rpt3 and mutations in the Rpt5 motif (Fig. 4d, Supplementary Data 2). The C-termini of Rpt1 and Rpt6, known to induce gate opening[40], are not visibly buried within the α-ring.

## Phylogeny and reductive evolution of microsporidian proteasomes

The taxonomic placement and organization of microsporidia has undergone significant revisions due to the absence or reduction of several organelles and the remarkable sequence variation between species. Indeed, two evolutionarily adjacent *Nematocida* spp. share only ~68% of their sequence[53], and even in complexes as universally conserved as ribosomes, yeast and microsporidian homologs share only 38% sequence identity on average[54]. This sequence diversity confounds not only phylogenetic classification, but also attempts to identify microsporidian homologs, which hampers our functional understanding of these ecologically important organisms. Several clear examples can be seen in previous work, where ribosomal and proteasomal proteins were unidentifiable using BLAST[31,55]. Structural studies revealed that several of these proteins are retained[23,25,28], albeit with low sequence conservation. In our own work, initial attempts to identify *V. necatrix* α subunits resulted in nearly identical top hits for α1, α4, and α5. Only by modeling the chains into density were we able to ascertain the correct sequences, reinforcing the importance of structural data as a complementary tool to assist in the annotation and functional characterization of microsporidian complexes.

We have identified microsporidian proteasome subunits on MicrosporidiaDB[56] and NCBI[44] using a combination of yeast sequences and structurally validated *V. necatrix* proteins as queries (Supplementary Data 2). The phylogenetic tree generated using high-confidence proteasomal subunits (Fig. 6a) is consistent with trees derived from simplified rRNA or whole-proteome analyses[54,57]. Saliently, a comparison of the relative length and conservation of proteasomal subunits between microsporidia and selected eukaryotes to *S. cerevisiae* reveals widespread evidence of reductive evolution (Fig. 6a), where nonessential subunits or sections are deleted to more efficiently utilize resources.

Although the functions of many eliminated fragments are not well-defined, some notable examples include the poorly resolved[58] C-terminal extensions from α3, α4, α5, and α7, which are associated with stabilization of CP-RP interactions, substrate recognition[59], nuclear localization[58], and recruitment of the quality control protein Ecm29[60]. The propeptides of the β subunits are also frequently modified. These segments are integral to the assembly process, where they protect catalytic residues and recruit other subunits and

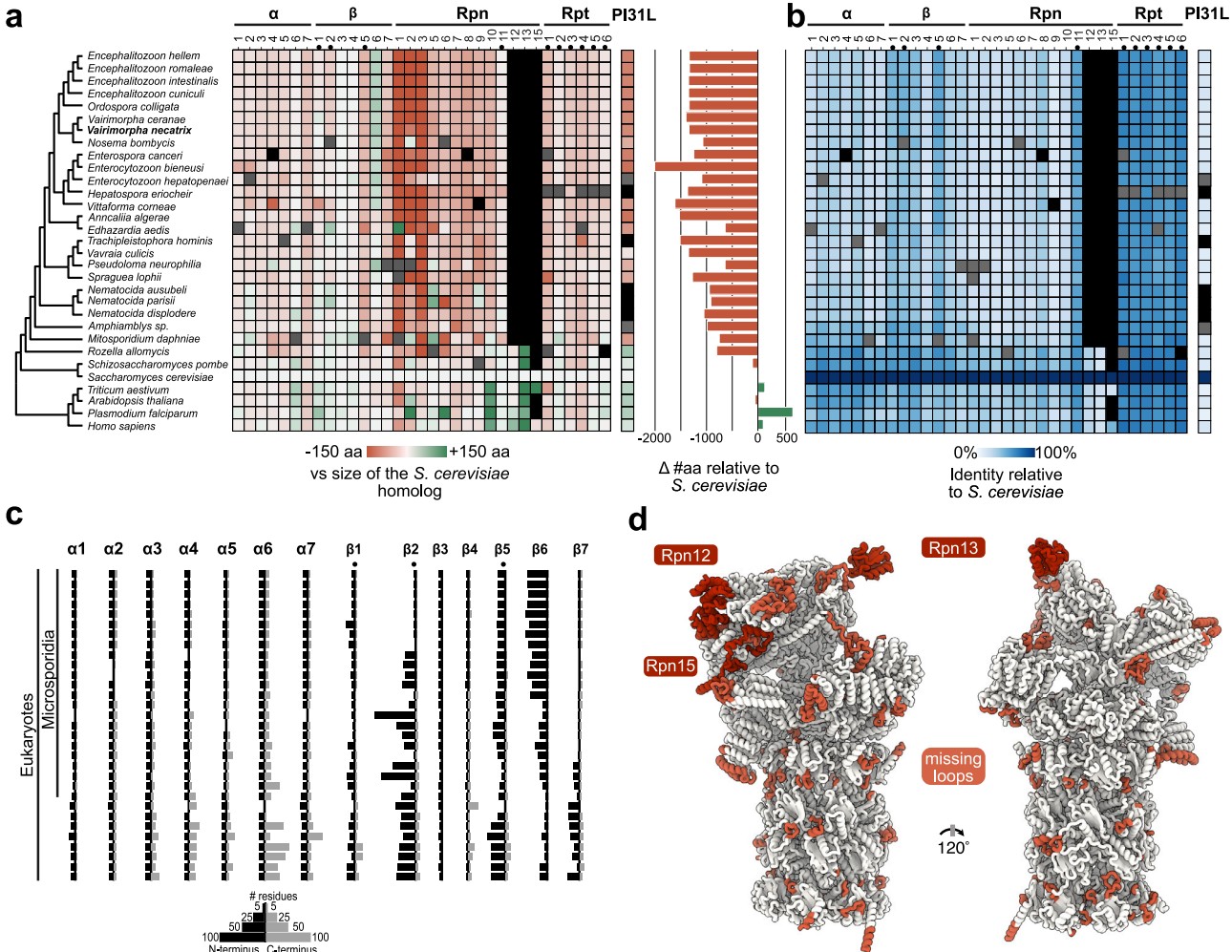

**Fig. 6 | Phylogeny and reductive evolution of the microsporidian proteasome.**
**a** Phylogenetic tree of proteasome core and regulatory subunit sequences of microsporidia and outgroups. The plot shows the sequence length of each protein (left) as well as the total proteasome sequence length (right) compared to *S. cerevisiae* (red: shorter, green: longer, gray: incomplete sequence, black: not clearly identifiable). PI31L proteins are depicted but were excluded from total length difference calculations. **b** Heat map of the protein sequence identity of each subunit and PI31 compared to *S. cerevisiae* (white: lowest sequence identity, dark blue:

highest sequence identity, gray: incomplete sequence, black: not clearly identifiable). **c** Histogram plots of the number of N- (black bars) and C-terminal residue (gray bars) from a defined position in the multiple sequence alignment of the individual subunits. **a-c** A black dot on top marks enzymatically active subunits. **d** Two views of the *S. cerevisiae* 26S structure (PDB-6FVU[40] [https://www.rcsb.org/structure/6FVU]) colored in white with the subunits Rpn12, Rpn13, and Rpn15, which are not identifiable in microsporidia, and missing internal and terminal elements colored in shades of red.

assembly factors[61,62]. The most consistent and sizable truncations occur within β5 and β7 propeptides (Fig. 6c), which recruit β6 and promote proteasome dimerization via interactions with Ump1, respectively[61,62]. Contrary to the tendency towards reduction, many microsporidians have expanded their β6 propeptides by >70 amino acids. These substantial changes to microsporidian propeptides suggest that the proteasome assembly pathway differs significantly from typical eukaryotes. Thus, while microsporidia express asymmetric proteasomes that are evolutionarily related to eukaryotes, the lengths and structural variability of α and β subunits are converging towards prokaryotic parameters. The average sequence lengths of *V. necatrix* α (233 ± 5) and mature β (204 ± 10) subunits are very similar to the lengths of α (233) and β (203) chains from the model archaeon *Thermoplasma acidophilum*. This is significantly smaller than the size and variability present in yeast α (257 ± 16) and mature β (214 ± 15) chains.

Microsporidia seem to have universally lost Rpn12, Rpn13, and possibly Rpn15, and almost all remaining proteins have been pared down, resulting in proteasomes ~1500 amino acids smaller than in yeast. Additionally, structural subunits display a marked decrease in

sequence conservation compared to enzymatic subunits (Fig. 6b). Large deletions are particularly common in the RP, with the Rpn1, Rpn2, and Rpn3 deletions exceeding 100 amino acids each.

In some cases, the significance of a deletion is readily apparent. For example, truncation of the ~80 C-terminal amino acids of Rpn2 coincides with the absence of Rpn13, known to bind to this region[63]. This observation is consistent with a previous hypothesis that the elimination of proteins in microsporidia has resulted in a simplified interaction network, which in turn enables the erasure of protein-interacting domains[64]. Likewise, the N-terminal ~100 amino acids of Rpn3 protrude from the proteasome lid in yeast and may stabilize the Rpn12 binding region. The loss of Rpn12 is thus accompanied by the loss of these Rpn3 residues in microsporidia. In Rpn1, deletions may denote regions where binding partners have yet to be identified. The ~100 amino acid deletion in Rpn1 connects two adjacent toroid domains in other eukaryotes but is absent from all reconstructions due to its structural heterogeneity[65]. Widespread conservation of this disordered segment in other eukaryotes signifies an important functional role, such as the potential recruitment of an unknown partner, which may be absent in microsporidia.

Although it is tempting to draw conclusions from the absence or extreme modification of several microsporidian orthologs, isolated outliers are most likely the result of incomplete genome assemblies and further work is necessary to establish the veracity of any remarkable modifications. On the other hand, alterations present in a cluster of organisms, like the expansion of Rpn6 in *M. daphinae* and *Nematocida* spp. (Fig. 6a), may be worth investigating to identify the potentially expanded repertoire of proteasome functionalities.

PI31-like proteins were identifiable in most microsporidian species using a combination of structural homology and sequence similarity searches at key interaction interfaces (Fig. 6, Supplementary Data 2). PI31L is truncated by nearly 40% in most microsporidians compared to yeast, and displays low sequence identity to yeast PI31 (Fig. 6a, b). Nonetheless, the identification of orthologs in most species suggests a common role for PI31L in the microsporidian lifecycle.

## Discussion

Stringent reductive evolution in microsporidia has eliminated many genes required by free-living species, including those necessary for sugar/fat metabolism, amino acid/nucleotide biosynthesis, and energy generation via oxidative phosphorylation[66]. Microsporidia are thus left with a core of only 800 conserved proteins, in addition to species and clade specific proteins[26]. Proteins that are retained are involved in vital cellular processes like protein synthesis and recycling[64]. The considerable reduction of microsporidian proteins and protein complexes makes microsporidia ideal organisms in which to study minimized versions of essential and highly conserved macromolecular assemblies. Previous work on the ribosome has demonstrated that microsporidia have rolled back many of the expansions characteristic of eukaryotic ribosomes and deleted several proteins known to be dispensable in yeast[23,25,28]. These data suggest that microsporidian complexes can be used to highlight regions with nonessential or expanded functionalities in eukaryotes.

The 20S structure described in this work is the most minimized eukaryotic proteasome studied to date and reveals that microsporidia have deleted many of the insertions that differentiate individual α and β subunits. Overall, the 26S proteasome is ~170 kDa smaller in *V. necatrix* than in yeast (Fig. 6), with nearly every subunit undergoing some level of reduction. Truncations most frequently occur in disordered loops or at the N- and C-termini. As short motifs used for the recognition of interaction partners often occur within disordered segments, the deletion of these regions results in a much more simplified proteasome interaction network when compared to other eukaryotes.

Unlike bacterial and archaeal proteasomes, which can self-assemble, eukaryotic proteasomes require a diverse group of assembly factors to assist in the formation of mature particles[67]. Core particle assembly in eukaryotes is largely mediated by Pba1-4, Blm10, Ump1, and the propeptides of the β subunits. As in a previous study[31], we were unable to identify any of these assembly factors in most microsporidians. This, in combination with the frequent changes to the propeptides, suggests that microsporidia utilize an unusual assembly pathway. Intriguingly, the β6 subunit is the only proteasomal protein expanded in most microsporidian species, due to extension of the propeptide (Fig. 6c). One possibility is that β6 has cannibalized the roles of the truncated subunits by incorporating assembly factor binding sites into the β6 propeptide, and the low sequence identity inherent to microsporidia has made identification of assembly factors difficult. Alternatively, microsporidian proteasomes may be amenable to self-assembly, like bacterial and archaeal equivalents, with β6 playing a larger role in the assembly process. These changes facilitate simplification of the proteasome interaction network, allowing microsporidia to conserve resources by eliminating unnecessary genes.

Proteasomes account for 5% of the total protein content in yeast[68]. They therefore represent a large investment of nutrients and require consistent ATP availability to function. Microsporidian spores display very little background metabolic activity and are still infective after a year in environmental conditions[69]. It is thus imperative for microsporidia to conserve energy and nutrients during the extracellular spore stage, necessitating a means to impede proteasome activity. Previous studies have confirmed that microsporidian spores are rich in proteasomal proteins, indicating they are not digested during the extracellular stage[30]; however, it was unknown how microsporidia regulate proteasomal activity.

We have identified PI31-like peptides in proteasomes isolated from microsporidian spores that occludes all six active sites. In yeast, PI31 binds to proteasomes to inhibit both proteolytic activity and the ATP-dependent formation of 26S proteasomes[9,10]. Notably, proteasomes purified from sporoplasms in the presence of ATP lack density for PI31L and occasionally associate into complete 26S and 30S proteasomes. These data suggest that a combination of factors contribute to the reactivation of proteasomes after hibernation. We therefore speculate that PI31L acts as a hibernation or dormancy factor in microsporidian spores. After microsporidia emerge from hibernation, PI31L is either cleaved or expelled, resulting in the eventual reactivation of proteolytic activity. Interestingly, this process may be reminiscent of the PSG-based storage pathways present in other eukaryotes during carbon starvation, where nutrients are conserved by sequestering proteasomes rather than degrading them[7,8].

Although PI31L likely represents an intentional strategy for the inhibition of proteasomes in dormant spores, the structure we have identified may also be used as a guide to produce targeted inhibitors of microsporidian proteasomes. Prior studies in microsporidia have suggested that they are particularly sensitive to proteotoxic stress due to limitations imposed by genome compaction[31]. In yeast, 20 amino acid fragments of PI31 have been used to selectively inhibit the β2 subunit active site[10]. PI31L is significantly shorter than yeast PI31 (Supplementary Data 2), and therefore represents an efficient alternative to simultaneously inhibit all three proteolytic active sites. The unique binding mode identified at the β1 active site suggests PI31L mimetics are a feasible option for potential therapeutics against these emerging pathogens. Additionally, the genomic streamlining in microsporidia has resulted in a highly differentiated and minimal version of eukaryotic proteasomes, revealing unique interaction interfaces that may be utilized in future studies to serve as drug targets.

## Methods

### Cultivation and isolation of *V. necatrix*

*V. necatrix* was cultivated and reproduced by feeding ~100,000 spores to fourth and fifth instar *Helicoverpa zea* larvae, grown on a defined diet (Benzon Research). After 3 weeks at 21–25 °C, the spores were harvested. First, larvae were homogenized in water, followed by filtration through two layers of cheesecloth and subsequent filtration through a 50 μm nylon mesh. The filtrate was layered on top of a 50% Percoll cushion in a 2-ml microcentrifuge tube, and spores were pelleted via centrifugation at 1000 g for 10 min. The pure spores were stored at −80 °C until further use.

### Purification of the *V. necatrix* proteasome

Proteasomes were purified from either dormant or germinated spores to identify differences between inactive and activated complexes. For spore-based preparations, 100 mg of *V. necatrix* spores were suspended in 500 μl size exclusion chromatography (SEC) buffer containing 50 mM Tris (pH 7.4) and 300 mM NaCl, supplemented with 5 mM DTT and a protease inhibitor cocktail (Complete EDTA-free, Roche). Spores were transferred to tubes containing Lysing Matrix E (MP Bio), and lysed via three, 1-min intervals in a Fast-Prep 24 (MP Bio) grinder at 5.5 m/s. Complete lysis was guaranteed by a following sonication step and verified via light microscopy. Spore debris was then pelleted via centrifugation for 20 min at 20,000 g, before loading the supernatant onto a Superose 6 Increase 10/300 SEC column

(Cytiva). Fractions containing proteasomes were identified using a combination of SDS-PAGE and negative stain EM.

To obtain sporoplasm-derived proteasomes, 5 mg of *V. necatrix* spores were germinated via alkaline priming[33]. Briefly, spores were incubated in 500 µl of 0.1 M KOH for 15 min at 22 °C, followed by pelleting via centrifugation at 10,000 g for 2 min. Primed spores were then germinated by resuspension in 650 µl of 0.17 mM KCl, 1 mM Tris-HCl (pH 8), and 10 mM EDTA. Germination was confirmed by light microscopy. Approximately 80% of spores germinated within 5 min; following which, 650 µl of rich insect cell media (ExCell 420) supplemented with 20 mM ATP, was added to the germination mix. Sporoplasms were incubated for 20 h at 22 °C, then pelleted via centrifugation at 10,000 g for 2 min and resuspended in 500 µl proteasome buffer containing 50 mM Tris (pH 7.4), 100 mM NaCl, 10 mM MgCl2, 20 mM ATP, 1 mM DTT, and 1% glycerol. Sporoplasms were then lysed via mild sonication (6 µm amplitude) for a total of 12 s, and lysis was validated using light microscopy. Purification of proteasomes proceeded via differential centrifugation in proteasome buffer, with sequential hour-long spins at 21,000 g, 54,000 g, and 121,000 g at 4 °C. Pellets were then resuspended in 50 µl proteasome buffer without glycerol. Negative stain EM and SDS-PAGE were used to verify 20S and 26S proteasome enrichment in the 121,000 g fraction.

## Phylogenetic analysis

The proteasome sequences were retrieved by tblastn searches from NCBI[44], MicrosporidiaDB[56], and an in-house *V. necatrix* genome (manuscript in preparation), using *S. cerevisiae* and *V. necatrix* sequences as queries with an E-value cutoff of 0.05. The phylogenetic tree was generated by protein sequence alignment using MUSCLE5 (v5.1)[70]. Alignments were trimmed using trimAl (v1.4.1)[71] with the −gappyout option. FASconCAT-G (v1.05.1)[72] was used to concatenate the trimmed alignments and to create the partition file. The phylogenetic tree was constructed using iqtree (v2.2.0)[73] with the -MFP MERGE function to identify the best partition model, followed by tree reconstruction using 1000 bootstrap replicates. The -p function was used to allow each partition to have its own evolution rate. Tree rooting was done in Figtree (http://tree.bio.ed.ac.uk/software/figtree/, v1.4.4). The sequence identity heatmap was constructed using MUSCLE5 (v5.1)[70] with *S. cerevisiae* sequences set as reference.

## Cryo-EM grid preparation and data collection

Purified proteasomes were applied as 3.5 µl aliquots to Quantifoil R2/1 200-mesh copper grids (EM sciences, Prod. No. Q250CR1) or R1.2/1.3 400-mesh gold grids with 2-nm carbon (EM sciences, Prod. No. Q450AR1.3-2nm), for spore and sporoplasm samples, respectively. Grids were glow discharged for 30 s at 15 mA before sample application. Grids were blotted for 5 s in an FEI Vitrobot Mark IV (Thermo Fisher Scientific), set to 4 °C and 100% humidity, prior to plunge-freezing into liquid ethane.

Cryo-EM data were collected at the Umeå Core Facility for Electron Microscopy. For 20S structures, data were collected with a pixel size of 1.042 Å on a Titan Krios (Thermo Fisher Scientific) operated at 300 kV using a Gatan K2 BioQuantum direct electron detector. Data for the 26S structure was collected at a 1.495 Å pixel size, using a 200 kV Glacios system (Thermo Fisher Scientific) equipped with a Falcon 4i direct electron detector. A total of four EPU data collections were used to generate the structures for spore-derived 20S, sporoplasm-derived 20S, and sporoplasm-derived 26S structures. Data collection statistics are summarized in Supplementary Table 1.

## Cryo-EM image processing

The four datasets were processed using a combination of Relion (v3.1)[74], Relion (v4.0)[75], and cryoSPARC (v3.3.2)[76]. The procedure is outlined in Supplementary Fig. 1. Briefly, in all datasets, movie alignments, drift correction, and dose weighting were done using

MotionCor2[77], and CTF estimation was performed with CTFFIND (v4.1.14)[78]. Micrographs with poor CTF fits or non-ideal ice thickness were removed, resulting in 3935 micrographs for the spore-derived dataset one, 3,416 micrographs for the spore-derived dataset two, 3277 micrographs for the sporoplasm-derived 20S data set, and 2,658 micrographs for the 26S dataset (Supplementary Table 1). For the two spore datasets, particles were first auto-picked in Relion (v3.1) using a Gaussian-blob template. Particles were classified in 2D, and promising classes were then used as templates for a second round of automated particle picking. Template-picked particles were extracted with a box size of 400 pixels (416 Å), filtered via iterative 2D and 3D classification, and further refined via per-particle CTF refinement and Bayesian Polishing in Relion (v3.1). Polished particles were then exported to cryoSPARC and homogeneously refined, with C2 symmetry imposed. Map resolution was determined by the Fourier shell correlation (FSC) between two half-maps at a value of 0.143, resulting in a nominal 20S resolution of 2.77 Å (52,679 particles).

Micrographs from the sporoplasm-derived samples were manually picked in Relion (v4.0). The 20S map was refined using the 1.042 Å pixel size dataset. Manually picked particles were extracted with a 400-pixel box size (416 Å) before exporting them to cryoSPARC for ab initio modeling and homogeneous refinement with imposed C2 symmetry, leading to a 3.2 Å map (17,942 particles). To offset the low 26S particle density, a 1.495 Å pixel size dataset was collected. Manually picked particles were again extracted with a box size of 400 pixels (598 Å), filtered via 3D classification, and non-uniformly refined without symmetry constraints. The final 26S map has a nominal resolution of 8.3 Å (6442 particles) at an FSC value of 0.143.

## Model building and refinement

A high-resolution yeast proteasome crystal structure (PDB-5CZ4[32] [https://www.rcsb.org/structure/5CZ4]) was used as an initial template for modeling *V. necatrix* proteasomes in Coot[79] (v.8.9.3). Model geometries and fits to maps were adjusted in ISOLDE[80] (v1.3). Final refinements were performed using PHENIX[81] (v1.20-4487) real space refinement against the final map, resolved to 2.8 Å. Model statistics are described in Supplementary Table 1, and model compositions are described in Supplementary Table 2. Sidechains in poorly resolved areas were removed due to insufficient data. Structures and maps were visualized and presented using ChimeraX[82].

To assess the *V. necatrix* 26S proteasome cryo-EM density, models of individual subunits were generated using AlphaFold[41,42] with the yeast structure (PDB 6FVU[40] [https://www.rcsb.org/structure/6FVU]) as a template. Top hits for each structure were then superimposed onto the yeast structure (PDB 6FVU[40] [https://www.rcsb.org/structure/6FVU]), which most closely fit to our 8.3 Å 26S proteasome map, and rigid body docked to best fit the available density.

## Reporting summary

Further information on research design is available in the Nature Portfolio Reporting Summary linked to this article.

## Data availability

The cryo-EM density maps, the coordinates, and the raw micrographs for the microsporidian proteasome have been deposited in the EM Data Bank (https://www.ebi.ac.uk/pdbe/emdb/), the Protein Data Bank (https://www.rcsb.org), and the Electron Microscopy Public Image Archive (https://www.ebi.ac.uk/pdbe/emdb/empiar/) with accession codes EMD-15365, PDB-8ADN, and EMPIAR-11117 for the spore stage 20S core-particle, EMD-15367 and EMPIAR-11118 for the sporoplasm-derived 20S core-particle (post-germination), and EMD-15366 and EMPIAR-11121 for the full 26S proteasome. The yeast proteasome structures (PDB-5CZ4[32] [https://www.rcsb.org/structure/5CZ4]) and (PDB 6FVU[40] [https://www.rcsb.org/structure/6FVU]) were used as templates.

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

## Acknowledgements

We thank B. Vossbrinck for her help in editing the manuscript, SV Pipaliya for phylogenetic advice, and members of the Barandun laboratory for discussions and critical reading of this manuscript. Further, we thank Michael Hall and Camilla Holmlund for help with cryo-EM data collection. The electron microscopy data was collected at the Umeå Core Facility for Electron Microscopy, a node of the Cryo-EM Swedish National Facility, funded by the Knut and Alice Wallenberg, Family Erling Persson and Kempe Foundations, SciLifeLab, Stockholm University and Umeå University. N.J. is supported by an Integrated Structural Biology fellowship from Kempe (JCK-1918). J.B. acknowledges funding from the Swedish Research Council (2019-02011), the European Research Council (ERC Starting Grant PolTube 948655), the SciLifeLab National Fellows program, and MIMS. C.R.V. acknowledges funding from the Hatch Grant Project CONH00786 and R. Tyler Huning. The computations were enabled by resources provided by the Swedish National Infrastructure for Computing (SNIC) at High-Performance Computing Center North (Project Nr. SNIC 2021/23-718 and SNIC 2021/22-936), partially funded by the Swedish Research Council through grant agreement no. 2018-05973.

## Author contributions

N.J. and J.B. conceived the study. C.R.V. cultivated microsporidia. N.J. purified proteasomes and performed all EM work. N.J. and K.E. determined the cryo-EM structures and together with R.R.W. built the structural model. R.R.W. and D.S. performed phylogenetic analysis and all authors interpreted the results, wrote, and edited the manuscript.

## Funding

## Competing interests

The authors declare no competing interests.
