## [Peer Review File · Nature Communications]

Structure of the reduced microsporidian proteasome bound by PI31-like peptides in dormant spores.Reviewers' comments:

Reviewer #1 (Remarks to the Author):

Nathan Jesperen et al. in this manuscript reports on a few cryo-EM structures of the microsporidian 20S and 26S proteasomes from dormant or germinated *Vairimorpha necatrix* spores. Based on the structural and phylogenetic analysis, the authors interpret the proteasome structure as “reductive evolution”. I found the data in this study are rather interesting albeit with quite a few unsolved issues and might provide more important insights if the authors can clarify the following questions, which may be necessary to reach a convincing conclusion. While the current structural data look valid, the lack of certain biochemical examination and analysis has stymied a necessary revelation of underlying mechanism. Therefore, a major revision appears to be necessary for this paper before it gets accepted for publication in Nature Communications.

Major issues:

The spore 20S proteasome appears to have all six active sites bound to non-proteolytic peptide, and thus fully inhibited. Please explicitly confirm and discuss this if it is true. This result immediately resembles a recent study just published in Nature Structural & Molecular Biology on the yeast PI31/Fub1-bound proteasome (<https://doi.org/10.1038/s41594-022-00808-5>). Therefore, the authors should conduct additional work to address the following issues:

- (1) What is the endogenous identity of the “non-proteolytic” peptides bound to the spore 20S proteasome active sites? Are they the ortholog of yeast PI31/Fub1? Please conduct additional biochemical analysis to clarify these. If it is true, the authors might have capture important evidence for the underappreciated biological function of PI31/Fub1, that is, PI31/Fub1 keeps the spore 20S proteasome in an inactive, dormant form.
- (2) It appears that the additional “non-proteolytic” peptide densities are of sufficient quality to allow side-chain fitting. The authors might want to conduct do novo atomic modelling to confirm the identity of these peptide densities.
- (3) Make necessary comparison to the PI31/Fub1-bound yeast 20S proteasome structure just published, so that we can better understand the nature of these “non-proteolytic” peptide.
- (4) Are there a portion of spore 20S particles not bound to the “non-proteolytic” peptide fragments?
- (5) If the above suggested analysis is of positive results, then the authors may need to propose a more relevant mechanism: (1) the PI31 ortholog in *Vairimorpha necatrix* inhibits the 20S proteasome in dormant spore. (2) Once germinated, the 20S proteasome is activated in the 26S form and cleaves or degrades PI31 ortholog to release them and fully activate itself. (3) Therefore, the PI31 ortholog inhibits 20S but much less so for 26S. In this regard, the observed “non-proteolytic” peptides are still proteasomal substrate as opposed to the so-called “non-proteasomal” in the text (line 751) but have very different degradation kinetic properties that endow its ability to inhibit 20S.

(6) If the above issues can be well resolved, the paper will then shift to a more interesting topic rather than focus on “reductive evolution”, which itself does not need a 20S structure to appreciate. Apparently, how the dormant spore 20S is inhibited (if true) and then fully activated in germinated spores, hypothetically by a PI31 ortholog or functionally similar protein, will be a more attractive, important and novel mechanism to reveal. However, such a mechanism is mostly not developed at all in the present form, while it looks like the authors already have most of the data for making such a development, with a few more minor experiments to consolidate it.

Minor issues:

(1) Please make necessary comparison with the human 26S proteasome in the resting and activated states (ref. 46). The potential importance of understanding parasite proteasome structures is to develop inhibitors that selectively inhibits the parasite 20S but not the human 20S/26S. Thus, the difference between the parasite and human structures are the most relevant information for inhibitor development.

(2) Figure 4d: the authors present a “partially open” CP gate. However, we know from the published studies (see Chen et al. PNAS 2016, 46, 12991-12996; for a thorough review on this, see https://doi.org/10.1007/978-3-030-58971-4_1 and the references therein) that 26S proteasome in the absence of substrate spontaneously samples both closed and open CP conformations. Therefore, the “partially open” CP gate could be simply averaged from the closed and open CP from different conformational states. In this regard, interpreting the 26S reconstruction as single “partially open” state will be misleading or a mistake. Please correct or replace it with appropriate discussion (between line 268-281).

(3) Line 90-91, please add “genetically” before “missing Rpn12, ...” or alike to avoid confusion.

(4) Citation formats are inconsistent and missing necessary information. Many are lack of the article number or the last page or page range. Please fix.

(5) Mismatch of citations: for example, ref. 75 on line 510 is meant to cite the reference of Coot, but the ref. 75 is another paper not directly explaining Coot. Refs 76-78 are completely missing in the section of References. Please correct.

Reviewer #2 (Remarks to the Author):

Microsporidia are eukaryotic parasites that have undergone genome compaction; as such their proteins typically display less complexity, as was recently shown in the structure of a microsporidial ribosome. Here, Jesperen et. al., solve the structure of the microsporidian proteasome using single particle cryo-EM, from dormant spores and germinated spores. The structure is interesting and shows how the

microsporidial proteasome, including the 20S core is reduced compared to other eukaryotic proteasomes. As discussed in detail below, it is less clear what conclusions can be drawn from solving the structure of the proteasome from germinated spores. We thank the authors for making their manuscript available to the community on BioRxiv!

Major Comments

The overall structure of the microsporidial proteasome, and comparison with other proteasomes is interesting. However, one of the major points of confusion for us is: what can be drawn from studying the structure of the proteasome from dormant spores Vs. spores that are germinated in vitro and incubated with ATP for 20 hours? Given that the germination process takes place on the timescale of milliseconds, and the interaction between the germinated spore and host cells will involve many other factors in addition to ATP, it was unclear to us exactly what question the authors were addressing in this experiment. For example, would lysing spores and incubating the resulting lysate with ATP produce the same results? Do the authors expect the process of germination to cause anything different? Or is the question simply to assess +ATP/-ATP conditions? If germination +ATP incubation is expected to result in something different, an important control would be to lyse dormant spores and incubate these with ATP. As currently presented, our understanding of the results are that this experiment may report primarily on +/- ATP conditions. We feel it would be important and helpful for the authors to clarify this, and either modify the analysis/conclusions based on existing data, or perform additional experiments that speak to the physiological implications of the dormant Vs. germinated samples that they have used in these experiments.

Minor Comments

1) We found some of the figures a bit challenging to follow

Figure 2 (a): Not immediately clear which the short fragments and long fragments are. It would be helpful to reader if marked explicitly on the figure

Figure 3 (a): It would be helpful if the position of S3 and S4 pockets more explicitly indicated

Figure 4 (a): From the figure itself and the legend it is unclear if we are looking at the N-terminal tail or C-terminal. It would be helpful to note this and mention it in the legend.

Figure 4b was confusing for us. It should show that a2 and a4 descend into the pore while a3 and a5 sit on the top. However, from the figure it looks like most of them are descending into the pore.

Figure 5 (a) and line 250: The concept of the grasping hand analogy is hard to clearly understand for a reader not familiar with proteasomes. Could this be more explicitly marked on the figure OR a schematic be added that helps guide the reader

Figure 6 (c): There is a mismatch between figure and legend. Figure shows N-ter is black and C-ter is grey. Legend states that N-ter is in grey and C-ter is in black

- 2) What is the % or number of particles that were actually 30S in the sporoplasm fraction? This should be noted
- 3) What are the peptides that they found in the 20S CP from spores? A little more discussion would be helpful. Are these expected to be host proteins or microsporidian proteins?
- 4) What is the role of microsporidia proteasome in sporoplasm/life cycle? This would be useful context for the reader to know
- 5) Please explain the method of fitting model into low resolution map
- 6) What outgroups were used to assess genome reduction? Is it branched from a clade in which other groups have it? It would be useful to refine the wording re. Evolution a bit more specifically and explicitly, since it is not necessarily true that more complex directly correlates to being more evolved

Reviewer #3 (Remarks to the Author):

In this manuscript Jespersen et al. present the first structural insights into the microsporidian proteasome. The cryo-EM structures of 20S and 26S proteasomes from dormant and germinated spores of *Vairimorpha necatrix* reveal numerous similarities to proteasomes of higher eukaryotes, but also show several differences that may suggest somewhat distinct mechanisms for microsporidian proteasome assembly, function, or activity. The authors propose a slightly different gating mechanism for the 20S core, a potentially different substrate specificity of the beta-2 tryptic site due to changes in the charge distribution of the S3/S4 specificity pockets, and a more open S1 pocket in the beta-5 chymotryptic site that resembles mammalian immunoproteasomes and may support higher-affinity binding of proteasome inhibitors like PR-957. Furthermore, the 20S proteasome purified in the absence of ATP from dormant spores shows additional peptide-like densities near the proteolytic active sites, which the authors interpret as potentially inhibitory peptides that either represent indigestible fragments from an abundant substrate or peptides highly expressed in the spore. These peptide densities are not observed in the 26S proteasomes purified in the presence of ATP from germinated spores.

The structure of the 26S proteasome complex confirmed the absence Rpn12, Rpn13, and Rpn15 (Sem1), as previously proposed based on genome analyses, and it identified or confirmed several subunit truncations, including those of Rpn2, Rpn3, and Rpt5, further supporting reductive evolution as a general principle for microsporidian macromolecular complexes.

Significant sequence deviations compared to eukaryotic orthologs had previously rendered some of the subunit annotations challenging, and this structural study helped distinguishing and identifying particular proteins, especially the alpha subunits 1, 4 and 5 of the 20S complex. Using these structurally validated *V. necatrix* proteins as queries for data-base searches of microsporidian proteasome subunits, the authors generated a phylogenetic tree that is consistent with trees previously derived from rRNA or whole-proteome analyses.

Although the presented studies provide a first glimpse into the structure of the microsporidian proteasome, they are to a large extent confirmatory of previous genome or proteome analyses and otherwise highly speculative regarding potential functional consequences of the observed differences compared to other eukaryotic proteasomes, without providing sufficient experimental evidence. In the absence of more detailed, higher resolution structural and/or functional data, there are no major new revelations that go significantly beyond of what was previously known or postulated and would justify publication in *Nature Communications*.

Major points:

1) Based on their proteasome purifications from dormant spores in the absence of ATP and from germinated spores in the presence of ATP, the authors propose that only the germinated sporoplasm contains assembled 26S proteasomes, which lack the extra peptide-like density within the 20S proteolytic chamber. However, in the absence of appropriate controls it is unclear whether the purified complexes indeed reflect the scenario present in dormant versus germinated spores. Based on the provided data, it cannot be ruled out that some of the proteasomes were assembled as 26S or 30S complexes within the dormant spore, despite the absence of ATP, but dissociated during the purification process. Conversely, the 12 % assembled 26S or 30S complexes isolated from germinated sporoplasm may have assembled during the purification in the presence of ATP. The authors even acknowledge “that either the germination or the presence of ATP in the buffer facilitated the association of the RP and CP”, but the latter possibility was neither tested nor ruled out. Importantly, there is no experimental evidence at all for the authors’ claim that their data indicate a rapid reactivation of proteasomes after host infection.

As controls to assess the presence of 26S or 30S proteasomes in dormant and germinated spores and whether the observed peptide fragments in the 20S chamber are indeed expelled upon RP binding, the authors should purify proteasomes from dormant spores in the presence of ATP and from germinated spores in the absence of ATP.

Importantly, independent of the purification conditions, the isolation by size-exclusion chromatography will not be suited for drawing any conclusions about the speed of proteasome reactivation after host infection, as claimed by the authors.

2) The authors propose that the additional densities observed in the catalytic chamber of 20S CP from dormant spores represent indigestible and potentially inhibitory peptide fragments that occlude proteolytic active sites by interacting with several of the substrate-binding cavities and avoiding direct contact with the S1 and S2 sites, with the peptide backbone tracing 4 - 5 Angstrom away from the

catalytic threonines. However, these claims are not well supported by the presented figures, and the authors should show in Fig. 2 the pockets and their occupancy by peptide side chains (or lack thereof) in greater detail, not just some blobs. The claimed 2.8 Angstrom resolution should certainly allow a more detailed depiction. Can it be ruled out that higher flexibility or non-identical sequences of the bound peptides cause the lack of density in some areas? To support their hypothesis that these peptides are present only in dormant spore proteasomes, the authors should assess whether 20S proteasomes purified from germinated sporoplasm in the absence of ATP show this extra density.

3) It is proposed that microsporidian 20S proteasomes use a different gate access mechanism. However, there appear to be much more similarities than differences compared for instance to the yeast *S. cerevisiae* proteasome. Although the microsporidian gate may use a short alpha-helical segment of alpha-5's N-terminus as a plug, while the N-terminus of yeast's alpha-5 is more unstructured, both gates primarily rely on the alpha-2 N-terminus for gate closure. Instead of (or in addition to) showing a superimposition of the *S. cerevisiae* open and *V. necatrix* closed gates in Fig. 4c, the authors should show a superimposition of both closed states, which are expected to look very similar. The authors' claim that the N-termini of alpha-1, 6, and 7 in the closed-state of microsporidian 20S resemble those in the open state of yeast 20S is not supported by the presented figures, which depict just low-resolution sausages for the subunit backbones. The authors also discuss Tyr7 and Pro15 of alpha-6, but none of these details are shown, neither in the main figures nor in the supplement.

4) The structure of the microsporidian 26S proteasome confirms the absence of Rpn12 and Rpn13, which was previously proposed based on genome analyses, and the authors also conclude that no ortholog of Rpn15/Sem1 is present. However, can the presence of a short Sem1-like peptide indeed be ruled out at an approximate resolution of > 15 Angstrom in this region of the proteasome lid? The very superficial depictions in Fig. 5 do not allow any assessment of this.

The authors also state that microsporidian Rpt5 lacks the N-loop from its N-terminal domain. Again, no details are shown, and the numbering (residues 107 – 144 in S.c. Rpt5) is incorrect. A structured N-loop has so far only been described for human Rpt5, where it comprises residues 99-119, which are equivalent to residues 88 – 114 in S.c. Rpt5. (And there is a typo in the PDB citation for the S.c. structure in figure 5, which needs to be 6FVU, not 6VFU.)

The authors propose that the absence of Rpn13 and Rpt5's N-loop would have functional consequences for ubiquitin binding and cleavage, yet this is pure speculation and there is no supporting experimental evidence. The functional importance of Rpt5's N-loop has never been investigated for any eukaryotic proteasome, and mutations of Rpn13 in yeast *S. cerevisiae* show no significant phenotype on their own.

The authors state that the absence of Rpn13 also results in the deletion of its binding partner UCH37, a deubiquitinase that serves a proofreading function for inadvertently ubiquitylated proteins at the proteasome. It is discussed that this is consistent with the hypothesis that editing and high-fidelity tasks are common targets for macromolecular streamlining in microsporidia. However, UCH37 is also lacking in *S. cerevisiae*, which seems perfectly fine without it.

5) Based on a comparison between six known conformational states of the yeast S.c. proteasome and quaternary subunit arrangements in the V.n. proteasome, the authors conclude that their structure is most similar to the s2 state. However, considering the great similarities between all conformations except for s1, this assessment appears not very reliable at a resolution of > 10 Angstrom for the ATPase

ring and > 15 Angstrom for the lid, and it is certainly not convincing in the absence of any detailed figure representation. Given that the microsporidian proteasome most likely samples different conformational states similar to other eukaryotic proteasomes, it seems unfounded for the authors to postulate that the V.n. proteasome adopts a primed, but inactive conformation whose incompletely closed 20S gate “may allow for a more passive degradation of substrates, allowing microsporidia to conserve vital ATP.”

6) Regarding the numerous missing loops and truncated termini for several microsporidian proteasome subunits, has this been confirmed by sequencing in all cases or is this primarily based on structural information, where higher flexibility and lower local resolution may obscure more extended or unstructured segments? There are no legends or descriptions provided for supplementary data 1 and 2, and it is unclear whether the sequences for the proteasome subunits listed in supplementary data 1 originate from mass spectrometry or genome sequencing.

7) The introduction mentions Rpn4 as a structural protein of the proteasome, which is incorrect.

Overall, this manuscript contains a lot of speculations and draws numerous conclusions with respect to microsporidian proteasome assembly and function that are not supported by experimental evidence. Besides providing the first low-resolution structures and a confirmation of a reductive evolution for microsporidian proteasomes that was previously proposed based on genome and proteome analyses, the manuscript lacks substantial and solid take-home messages. It appears premature and is therefore not suited for publication in Nature Communications.

Response to reviewers

We want to thank all three reviewers for the critical reading of our manuscript and the constructive assessment. Using inputs from all reviewers and a just published study highlighted by reviewer #1 (<https://www.nature.com/articles/s41594-022-00808-5.pdf>), we have reassessed our data carefully and can now unambiguously identify the active site peptide density as a distant homolog of the recently characterized proteasome inhibitor, PI31, bound to the microsporidian proteasome.

The microsporidian PI31-like (PI31L) protein is remarkably divergent from the yeast ortholog in sequence and structure. Protein identification, therefore, required a combination of (1) a multistep structure-based AlphaFold/Foldseek database search, (2) mass spectrometry data, and (3) excellent density/model fit.

The structural description of the microsporidian PI31L expands the importance and impact of the study and changes the take-home message considerably by

- identifying a novel proteasome inhibition mechanism in the dormant spore stage of microsporidia.
- providing the first PI31-like protein structure in a wildtype proteasome isolated from endogenous sources from an important pathogen.
- providing the structure and revealing differences of the minimal version of PI31L (147 amino acids in *V. necatrix*) compared to yeast (250 amino acids in *S. cerevisiae*).
- describing an ideal structural template for future design of microsporidian proteasome inhibitors

We have therefore made some significant changes to our study:

- The title has been changed to “***Structure of the reduced microsporidian proteasome bound by PI31-like peptides in dormant spores.***” to highlight the new take home message.
- The current state of research on PI31 is introduced, and the Introduction is modified and restructured to emphasize the significance of PI31.
- The Results and Discussion sections have been altered to highlight the unique arrangement displayed by PI31L in *V. necatrix*, as well as the role PI31L may play in the microsporidian lifecycle.
- PDB models have been updated with two chains of the microsporidian PI31L and we have made all cryo-EM raw data available on EMPIAR.

In addition to the major changes made to the manuscript as a result of the identification and structural description of the microsporidian PI31L version, we have addressed the reviewers’ comments point by point below and believe that the manuscript has strongly benefited from these changes.

Reviewers' comments:

Reviewer #1 (Remarks to the Author):

Nathan Jesperen et al. in this manuscript reports on a few cryo-EM structures of the microsporidian 20S and 26S proteasomes from dormant or germinated *Vairimorpha necatrix* spores. Based on the structural and phylogenetic analysis, the authors interpret the proteasome structure as “reductive evolution”. I found the data in this study are rather interesting albeit with quite a few unsolved issues and might provide more important insights if the authors can clarify the following questions, which may be necessary to reach a convincing conclusion. While the current structural data look valid, the lack of certain biochemical examination and analysis has stymied a necessary revelation of underlying mechanism. Therefore, a major revision appears to be necessary for this paper before it gets accepted for publication in Nature Communications.

We thank reviewer #1 for the positive assessment of our structural work, the careful review of our study, and particularly the very helpful input regarding the unidentified peptide-like density. These suggestions led us (1) to carefully reanalyse the data, (2) unambiguously identify and annotate the divergent microsporidian PI31-like (PI31L) protein, and (3) adjust text and figures to describe the microsporidian version of PI31 and its potential function to reduce protein degradation during the dormant spore stage.

Major issues:

The spore 20S proteasome appears to have all six active sites bound to non-proteolytic peptide, and thus fully inhibited. Please explicitly confirm and discuss this if it is true. This result immediately resembles a recent study just published in Nature Structural & Molecular Biology on the yeast PI31/Fub1-bound proteasome (<https://doi.org/10.1038/s41594-022-00808-5>). Therefore, the authors should conduct additional work to address the following issues:

(1) What is the endogenous identity of the “non-proteolytic” peptides bound to the spore 20S proteasome active sites? Are they the ortholog of yeast PI31/Fub1? Please conduct additional biochemical analysis to clarify these. If it is true, the authors might have capture important evidence for the underappreciated biological function of PI31/Fub1, that is, PI31/Fub1 keeps the spore 20S proteasome in an inactive, dormant form.

Although we had previously attempted to use a density-informed motif search of the *V. necatrix* proteome and mass spectrometry list to identify the protein bound to the active sites, we were not able to unambiguously assign the peptide-like densities. Your helpful observation, as well as the recent publication of the yeast PI31:proteasome structure, provided a starting point to carefully reanalyse our data.

Using a combination of HMMSearch and AlphaFold/Foldseek searches, we were able to identify divergent PI31 homologs in *V. necatrix* and other microsporidia (described in the new **Supplementary Figure 3**). The identified PI31L protein is significantly enriched in the spore-derived sample based on mass spectrometry data, where coverage for PI31L is higher than many proteasomal subunits (**Supplementary Table 1**, updated with PI31L annotation).

Density-Verified Proteasome Subunit	Top Yeast Hit	Yeast Accession	BLAST Evalue	Identification Exp. q-value	Coverage [%]	#PSMs	#Unique Peptides	Sequence
PSA2	PSA2	P23639	3,09E-33	0	85	108	23	MFLDTHKQLTFTSEGLDQCDAALKAATQSSLSVGVSDGVVLSAKESNLVILSEYKIQSPNLIQITVYSGCPDRIQVNLKISEYDITSTNIRLIRLVEFQSRQIYTIKKGYRPFGLTLLIV
PSA5	PSA5	P23639	3,09E-33	0	85	108	23	MFLDTHKQLTFTSEGLDQCDAALKAATQSSLSVGVSDGVVLSAKESNLVILSEYKIQSPNLIQITVYSGCPDRIQVNLKISEYDITSTNIRLIRLVEFQSRQIYTIKKGYRPFGLTLLIV
PSA7	PSA7	P21242	1,45E-24	0	81	79	16	MAILDVIDVYTTGDIQIGYAGADNGTACMKNNKGLMIAEKPIESKFNRIKVVNSIFGSSGIIETDLVYLNENKLNISEKHNDMDVHSRNVGIIHIFQTRYSVGRVIGIINL
PSB3	PSB3	P25451	1,86E-42	0	76	54	15	MSDISHYGSLLAMIGKSAFVLSLQKISSGVISVKNFTKYSLTPRFFFTGLVSDGEMFKIRKRVNQLVQDNNKDMPELSESNMISYLQKRLQVPIVSGVMTLQKPKYASSMDICGAI
PSB5	PSB5	P30656	1,38E-81	0	71	46	10	MEKLFITGIMELQITKVDASFMKNKIVPYKGTTLTAFIFQGGMVAVDSRASAGSYASQNVKRVKRVKHLIGTMAGGASDCYFEWKKMGLYAKLYELKNKRISVSAASMLNCKVYSYKGGGLS
PSA3	PSA3	P23638	2,12E-41	0	68	54	17	MNIFLLSLSLATNPDQDQEBTNTSNKSDENATESNLNQPNDSSVNPTELSDNNFANEQKEARNENPGLNNGNPEVVEGVLNADQKVFEGQVSLPQSSKNQGTGEGQAVDVSQKCSI
PSB7	PSB7	P30657	9,73E-27	0	66	39	13	MSAPGRGKSSKAGIAKRRHTSNEESIKPIAIRLARRAGVSRVSGSGCFNEHVAAKSYMTIIGISYFYANHARRKRTITSDVLSLKMGMKYMGMGY
PSA4	PSA4	P21242	6,17E-24	0	64	56	14	MEYENALGIFPDGRLQYVADQASQSSLVVSSDTEICLSIETKHNKMLIDQNKLLVDKLDNIWYITSGIKPDSYKLNARLICRINKYKNTITSFDELAYELSLVKKQFLDSSMRPFGRISIL
PSB4	PSB4	P22141	8,02E-26	0	59	37	13	MESSVALKNDPVIIGDSSVNSVYLKREEDKFNINNKVFTYLDGQDGFARTSFINKEKLYEEIQNVEITPVTANVCIQTLVDNLRSHPKNCYLVGGLSDQGPVLSYDGLSHENDFMVAI
PS46	PS46	P40302	4,66E-23	0	55	54	14	MSYDQYDYLKDDPKLVKNDHYLHNNKLTLETPDNYMIEQINDLVSYDYSYDRNLELGSNNKILGILNGENVEDDHPQGRKRRKGGKAGLFSFEEKEDSDGDKTDFLPLKRR
PI31L	PI31L	N.I.		0	54	12	6	MFQDQYDYLKDDPKLVKNDHYLHNNKLTLETPDNYMIEQINDLVSYDYSYDRNLELGSNNKILGILNGENVEDDHPQGRKRRKGGKAGLFSFEEKEDSDGDKTDFLPLKRR
PI31L	PI31L	N.I.		0	53	13	4	MNLLIFVSICTGITEVDPDSIEVAKIQQKIPPPDQQRILFAGKQLEDGRLTNDYNIQKESTLHLVRLRGGY
CDCA8	P25694	0	0	51	61	35		MQTNHEENKVDLSTALENTKALVDLHDHNEKLVQEVGNPQTEDELDLIGDYTLKGGKCEVFFLEMMNEPKTKICDGRVNLKRLINDVYKVPVCGSIVGELVPLIADTVEIKIGDLI
RL6B	P05739	4,36E-12	0	51	12	7		MIKDIQIRLQETAYQSDVPKYKEMKLNKQERKRRDTLDCGMVVCVGEYKRVYKLGDKNLCAVLKDGNNVFFKIDESYLATSTKININVNEHISEDEVPFLTKDKSIEIMDIENSEK
UAP1	P43123	1,95E-42	0	51	24	15		MEVNSNPKLKIYDNEGNLTEQGENYRKLGLSTLSCNFKGVVLSGGQSTRGCDGPGKFLKICGLTFEHHKHNKISNEYSNMILFIMTSDNTHEDVAVFKSNNFQDLVFFKQZSLTCTYENC
PSA1	PSA4	P40303	1,77E-34	0	47	52	11	MNLLIFVSICTGITEVDPDSIEVAKIQQKIPPPDQQRILFAGKQLEDGRLTNDYNIQKESTLHLVRLRGGY
PSA1	EAF3	Q12432	0,011	0	46	18	10	MDEKTDNDEDFLVEKLVNKKINGKPHYLKWDVYPSSENTWEPYDNDADDLKKYEEKIAKKSALQESKPGESITDSDQDKSKKETSLSANSTIKDVKDEKFDLTKQEKSSKSLKENEI
PSA1	N.I.		0	46	22	9		MRKISHVPTACLLVLFNIFRANANRQGPVLYTNGANGGAPVVDKPLVVPVNPCTKFKYNGRVRANTNPEQCAKADQKALRQALNAVAQKIVPEASPKPEECIRTVSNAGQCLK
PSA1	N.I.		0	45	10	9		MFISFRDLKTEASLILFDKQEQALTIPTKFRPYADPLDSTGEADALVGEYSERLKLSHYFFKQNEEFKAWVNSGDFEMNDTDKIKLNLVHVSXKDLPELFFTEGFLRASISGIRKLG
PSA1	RL26B	P53221	2,71E-20	0	44	10	7	MLAGRDKTKSLRKRKALFTAVGKDRKRMSSHLEELRSQVGFVSPVAVGDIKLVHSGYKQEGQLVFLVQDFRMTDRIYFEGLEENEEKVPAKLPCHVTFVFKFMNDMGREKQLKQEKIARVQ
PSA1	TCPA	P12612	1,16E-135	0	44	29	20	MQNEISNTIITGKSWKSGVAEKMTKSLVYNNKSSGFMGLDKMLNNAAGDVNITNDGATILQNMVIDDPAKILDLANTQDKEVGGTGLVLAACLSIEKMQNGMSPVWVNGY
PSA1	HZA1	P04911	4,22E-12	0	43	17	6	MATKGGKDPQSQADHTEGSSAVFKSSKIKKILKSKQRISQDACKAVSVAVLYMSEIMD GARNTANADGKIKLPHINSAICDCEVNLHGHMIIWIKSGVGSNAPLQELNKSD
PSA1	RS20	P38701	2,82E-07	0	42	5	4	MQVEKDDLEPQIHEMMKVVHIDISTNKSALACEKFAEAFENFESLKEVLPKQGLVPLTKPCGGGTITWAKYKIVHYZRYVTSDDQLKVKIQDFLKNPFINILGYTN
PSB1	PSB1	P38624	9,07E-56	0	41	32	9	MVAYDNKNNKSPNPEMFGTMMAVYAGIDGLADQICSTSMITVYSRFDLTKISDNVCCRSASADTQATCTITELVGRSSFDIKRPSKAAAKADIVPPMMLAGLIIAGIDYTESPFI

Screengrab of updated Supplementary Table 1. The updated annotation and sequence of PI31L is indicated in purple. Proteasomal subunits are colored green.

Further, the peptide densities are of sufficient quality to allow reliable side-chain fitting (new **Figure 2** and **Supplementary Figure 3**), resulting in a clear model to map fit. Taken together, we can unambiguously identify the peptide-like density as the microsporidian version of PI31 in *V. necatrix*.

(2) It appears that the additional “non-proteolytic” peptide densities are of sufficient quality to allow side-chain fitting. The authors might want to conduct do novo atomic modelling to confirm the identity of these peptide densities.

We have now built the structure of PI31L *de novo* and confirmed the identity of the protein. See response to point (1). Two additional protein chains have been added to the PDB file (chains 3 and 4) and **Supplementary Table 2**, which contains model composition and sequences. The PDB entry has also been updated accordingly.

(3) Make necessary comparison to the PI31/Fub1-bound yeast 20S proteasome structure just published, so that we can better understand the nature of these “non-proteolytic” peptide.

We have analyzed this interaction in detail (Lines 85ff, 158-196) and modified **Figure 2** to describe the overall structure of the two PI31L peptides and compare their fold and mode of action to the just published PI31/Fub1 structure from yeast.

Interestingly, the microsporidian version of PI31 is of much smaller size, has very low sequence identity with other PI31 orthologs, and differs in some key interaction regions significantly. We describe the structure, similarities and differences in an updated section (*Proteasomes from dormant spores are bound by PI31-like peptides*). Further, PI31 has been added to an updated **Figure 6** to highlight its reduced size and low sequence identity.

(4) Are there a portion of spore 20S particles not bound to the “non-proteolytic” peptide fragments?

We believe that a significant proportion of the particles used to reconstruct the final 20S structure from spores are bound by PI31L, and base our assumption on the following points:

- A PI31L-mask focused classification in Relion on the final particle set without image alignment and using C1-symmetry did not yield a class without additional density in the catalytic chamber (see the following Figure “PI31L-focused classification”). This suggests that the majority of the selected final particles are bound by the peptide, or the signal of the region within the mask is not strong enough for classification.
- Refinements with C2 symmetry as well as without symmetry (C1) result in essentially identical density features and intensities, suggesting similar occupancy at both binding sites.
- Mass spectrometry analysis shows the coverage of PI31L higher than many proteasomal subunits (see **Supplementary Data 1**) in the sample used for the cryo-EM grid preparation.

Spore 20S final particle set

Figure PI31L-focused classification. Overview of a PI31L-focused classification (yellow transparent mask) performed in Relion without image alignment and 3 classes. The 20S proteasome is shown on the left in light blue with the PI31L-densities in red. The final classes of PI31L are shown on the right, superimposed with the used mask in two 90-degree related views.

(5) If the above suggested analysis is of positive results, then the authors may need to propose a more relevant mechanism: (1) the PI31 ortholog in *Vairimorpha necatrix* inhibits the 20S proteasome in dormant spore. (2) Once germinated, the 20S proteasome is activated in the 26S form and cleaves or degrades PI31 ortholog to release them and fully activate itself. (3) Therefore, the PI31 ortholog inhibits 20S but much less so for 26S. In this regard, the observed “non-proteolytic” peptides are still proteasomal substrate as opposed to the so-called “non-proteasomal” in the text (line 751) but have very different degradation kinetic properties that endow its ability to inhibit 20S.

We thank the reviewer for the helpful analysis, and have incorporated a similar hypothesis on the mechanism of action for PI31L into the text (Lines 85ff, 425ff). We have additionally clarified that the active site peptide sections of PI31L in the observed state are in a conformation incompatible with digestion, but that does not mean they are indigestible.

(6) If the above issues can be well resolved, the paper will then shift to a more interesting topic rather than focus on “reductive evolution”, which itself does not need a 20S structure to appreciate. Apparently, how the dormant spore 20S is inhibited (if true) and then fully activated in germinated spores, hypothetically by a PI31 ortholog or functionally similar protein, will be a more attractive, important and novel mechanism to reveal. However, such a mechanism is mostly not developed at all in the present form, while it looks like the authors already have most of the data for making such a development, with a few more minor experiments to consolidate it.

We agree that the identification of the microsporidian version of PI31 provides a very interesting additional topic and have significantly altered the focus of our manuscript to highlight this novel mechanism.

Minor issues:

(1) Please make necessary comparison with the human 26S proteasome in the resting and activated states (ref. 46). The potential importance of understanding parasite proteasome structures is to develop inhibitors that selectively inhibits the parasite 20S but not the human 20S/26S. Thus, the difference between the parasite and human structures are the most relevant information for inhibitor development.

We are indeed interested in describing unique aspects of microsporidian proteasome structure to aid in the development of targeted inhibitors. Unfortunately, the resolution of our 26S structure is insufficient to achieve the atomic-level information necessary to distinguish enzymatic differences in the *V. nectatrix* regulatory particle and human proteasomes. In the current form, we focus on the conformational state of the $\beta 5$ active site methionine, which is associated with selective inhibition of immunoproteasomes in humans.

Most importantly, with the identification and structural description of PI31L, we now provide relevant information for the future development of microsporidian proteasome inhibitors. We have expanded on the utility of PI31L as a source for potential selective inhibitors (Lines 435-445), and describe its structure and major differences to yeast PI31 (Lines 85-90, 158-196).

(2) Figure 4d: the authors present a “partially open” CP gate. However, we know from the published studies (see Chen et al. PNAS 2016, 46, 12991-12996; for a thorough review on this, see https://doi.org/10.1007/978-3-030-58971-4_1 and the references therein) that 26S proteasome in the absence of substrate spontaneously samples both closed and open CP conformations. Therefore, the “partially open” CP gate could be simply averaged from the closed and open CP from different conformational states. In this regard, interpreting the 26S reconstruction as single “partially open” state will be misleading or a mistake. Please correct or replace it with appropriate discussion (between line 268-281).

We have adjusted the text to emphasize the conformational diversity that is likely present in our sample (Lines 308-314). We have also removed speculation as to the functional significance of the “partially opened” CP gate in the 26S proteasome and refer to it as an “intermediate state” in the updated **Figure 4d**.

(3) Line 90-91, please add “genetically” before “missing Rpn12, ...” or alike to avoid confusion.

We have clarified that these proteins appear to be absent at the genomic level (Lines 96-97).

(4) Citation formats are inconsistent and missing necessary information. Many are lack of the article number or the last page or page range. Please fix.

(5) Mismatch of citations: for example, ref. 75 on line 510 is meant to cite the reference of Coot, but the ref. 75 is another paper not directly explaining Coot. Refs 76-78 are completely missing in the section of References. Please correct.

We apologize for these mistakes and have corrected issues with citations.

Reviewer #2 (Remarks to the Author):

Microsporidia are eukaryotic parasites that have undergone genome compaction; as such their proteins typically display less complexity, as was recently shown in the structure of a microsporidial ribosome. Here, Jesperen et. al., solve the structure of the microsporidian proteasome using single particle cryo-EM, from dormant spores and germinated spores. The structure is interesting and shows how the microsporidial proteasome, including the 20S core is reduced compared to other eukaryotic proteasomes. As discussed in detail below, it is less clear what conclusions can be drawn from solving the structure of the proteasome from germinated spores. We thank the authors for making their manuscript available to the community on BioRxiv!

We thank reviewer #2 for the positive assessment of the structural work and the interest in the analyses of the reductive evolution of the microsporidian proteasome.

Major Comments

The overall structure of the microsporidial proteasome, and comparison with other proteasomes is interesting. However, one of the major points of confusion for us is: what can be drawn from studying the structure of the proteasome from dormant spores Vs. spores that are germinated in vitro and incubated with ATP for 20 hours? Given that the germination process takes place on the timescale of milliseconds, and the interaction between the germinated spore and host cells will involve many other factors in addition to ATP, it was unclear to us exactly what question the authors were addressing in this experiment.

For example, would lysing spores and incubating the resulting lysate with ATP produce the same results? Do the authors expect the process of germination to cause anything different? Or is the question simply to assess +ATP/-ATP conditions? If germination +ATP incubation is expected to result in something different, an important control would be to lyse dormant spores and incubate these with ATP. As currently presented, our understanding of the results are that this experiment may report primarily on +/- ATP conditions. We feel it would be important and helpful for the authors to clarify this, and either modify the analysis/conclusions based on existing data, or perform additional experiments that speak to the physiological implications of the dormant Vs. germinated samples that they have used in these experiments.

In the course of this revision and particularly based on the input from all reviewers and just-published complementary data (PMID: 35927584), we have carefully reanalysed the cryo-EM density and complementary data and can now unambiguously identify the unique peptide-like densities in the active sites as the microsporidian homolog of PI31/Fub1/PMSF.

The structural description of the microsporidian PI31L peptide now expands the importance and impact of the study and changes the take-home message considerably by identifying a novel proteasome inhibition mechanism in the dormant spore-stage of microsporidia. The successful identification of the peptide densities also allows us to address issues arising from our variable biochemical preparations. Previous work demonstrates that PI31 selectively inhibits 20S proteasomes (PMID: 25332237) and inhibits the ATP-dependent formation of 26S proteasomes (PMID: 24770418), suggesting a combination of factors contribute to the reactivation of microsporidian proteasomes after hibernation. We have modified the text significantly in response to your recommendations, and hope that the combination of text changes and background data on PI31 clarify the importance of both ATP and germination for the removal of PI31L.

Minor Comments

1) We found some of the figures a bit challenging to follow

Figure 2 (a): Not immediately clear which the short fragments and long fragments are. It would be helpful to reader if marked explicitly on the figure

Due to the identification of PI31L within the active site, we have now modified **Figure 2** extensively and hope that by adding specific labels, dashed lines for disordered regions and connections as well as specific colors for individual β -subunits, the location and arrangement of the peptides is now clearer. Further, we have added a sequence alignment of selected eukaryotic PI31 with all identified microsporidian PI31-like homologs to this figure. Fragments and their binding interfaces are labeled above this new sequence alignment and are also demarcated within the density in the same figure.

Figure 3 (a): It would be helpful if the position of S3 and S4 pockets more explicitly indicated

We have now marked the S3/4 binding pockets on the figure so that readers can more efficiently identify those binding interfaces. A closer view of these sites can also now be seen in the updated **Figure 2** and the new **Supplementary Figure 3**, where S3 and S4 sites are bound by the P3 and P4 residues of PI31L in each active site.

Figure 4 (a): From the figure itself and the legend it is unclear if we are looking at the N-terminal tail or C-terminal. It would be helpful to note this and mention it in the legend.

We have updated the figure legend to note that the sequence alignment focuses on the N-termini of the α subunits (Line 820), and emphasized this in the figure (**Fig. 4a**).

Figure 4b was confusing for us. It should show that $\alpha 2$ and $\alpha 4$ descend into the pore while $\alpha 3$ and $\alpha 5$ sit on the top. However, from the figure it looks like most of them are descending into the pore.

We apologize, the text annotation was incorrect and this should have referenced the closed yeast conformation in Figure 4c, where $\alpha 3$ and $\alpha 5$ sit on top of $\alpha 2$ and $\alpha 4$. We have corrected this mistake

(Line 245) and modified **Figure 4c** to include a side view, which should more effectively display the path of the N-termini with respect to the pore.

Figure 5 (a) and line 250: The concept of the grasping hand analogy is hard to clearly understand for a reader not familiar with proteasomes. Could this be more explicitly marked on the figure OR a schematic be added that helps guide the reader

We have added a small schematic and labels to the figure to assist in the identification of the ‘fingers’ in the grasping hand analogy.

Figure 6 (c): There is a mismatch between figure and legend. Figure shows N-ter is black and C-ter is grey. Legend states that N-ter is in grey and C-ter is in black

We thank the reviewer for the careful reading of our paper and have updated the figure legend accordingly (Line 854).

2) What is the % or number of particles that were actually 30S in the sporoplasm fraction? This should be noted

We have adjusted the text to note that about 1.5% of all proteasomal particles were capped on both ends (Line 266).

3) What are the peptides that they found in the 20S CP from spores? A little more discussion would be helpful. Are these expected to be host proteins or microsporidian proteins?

We have identified the peptide density as a microsporidian ortholog to the known proteasomal inhibitor PI31 and have significantly altered the text in response to this finding. Discussion of the binding and significance can be found in Lines 85-90, 158-196, 425-445.

4) What is the role of microsporidia proteasome in sporoplasm/life cycle? This would be useful context for the reader to know

We thank the reviewer for their interest in this subject, and we also find it to be a very intriguing subject! Very little is currently known about proteasome functionality throughout the microsporidian life cycle. In other organisms like yeast, proteasomes are known to be inactivated in nutrient-poor environments and reactivated after return to sufficiently rich environments, see, for example, commentary on PSGs (Lines 54, 433). There is even some fascinating work suggesting that bears temporarily inhibit proteasomes during hibernation to avoid muscle atrophy (PMID: 29615761). In microsporidia, it may be necessary to protect proteins from background digestion in the spore. Once they germinate and infect the host, microsporidia often quickly reproduce. In *Nematocida parisii*, the population doubling time is under 2.5 hours (PMID: 27782144). Proteasomes may play a vital role in the preparation for this rapid reproduction by degrading unnecessary proteins to generate amino acids for translation needs. Unfortunately, this is all speculation, as no studies have been performed to describe proteolytic activity at different timepoints in the life cycle. In light of this, we were reticent to comment on the role proteasomes might play after germination, and have therefore not included it in the text.

5) Please explain the method of fitting model into low resolution map

We have added a methods section to describe the fitting of our AlphaFold models into the density (Lines 543-546). Briefly, template-based AlphaFold models were superimposed onto the closest fitting yeast 26S structure, and then rigid body docked to most accurately fit the available density.

6) What outgroups were used to assess genome reduction? Is it branched from a clade in which other groups have it? It would be useful to refine the wording re. Evolution a bit more specifically and explicitly, since it is not necessarily true that more complex directly correlates to being more evolved

We appreciate the reviewer's interest in reductive evolution, and agree that the description of evolution favoring increased complexity was misleading. In fact, what we had intended to suggest is that although it is common for people to think of evolution as a march towards complexity, many organisms eliminate nonessential proteins or traits to conserve nutrients, increase replication rates, etc. Microsporidia offer a nearly unique glimpse into this process, as they have retained many macromolecular complexes that are typical to eukaryotes, but have often simplified these complexes and eliminated nonessential subunits (PMID: 35543997). We have modified the introduction to be more consistent with our focus on PI31 and in the process removed this discussion of evolution; however, reductive evolution is still an important theme in the phylogenetics section of this paper.

As you note, the point allowing us to differentiate reductive evolution from branching before an insertion is whether or not a given segment or protein is mandatorily present in a common ancestral species. The absence of that protein within microsporidians therefore represents an instance of reductive evolution, rather than an insertion in other clades. As an example, we show in Figure 6 that Rpn12 is present in fungi (yeast), plants (*A. thaliana*), animals (humans), and protozoans (*P. falciparum*). The presence of Rpn12 in all of these kingdoms suggests that Rpn12 existed in an ancestor of microsporidia, and must have therefore been eliminated at some stage in the evolutionary process. In this particular case, Rpn12 is even present in *Rozella allomyces*, a comparatively closely related species, indicating that Rpn12 has been lost after microsporidia and *Rozella* spp. branched.

Reviewer #3 (Remarks to the Author):

In this manuscript Jespersen et al. present the first structural insights into the microsporidian proteasome. The cryo-EM structures of 20S and 26S proteasomes from dormant and germinated spores of *Vairimorpha necatrix* reveal numerous similarities to proteasomes of higher eukaryotes, but also show several differences that may suggest somewhat distinct mechanisms for microsporidian proteasome assembly, function, or activity. The authors propose a slightly different gating mechanism for the 20S core, a potentially different substrate specificity of the beta-2 tryptic site due to changes in the charge distribution of the S3/S4 specificity pockets, and a more open S1 pocket in the beta-5 chymotryptic site that resembles mammalian immunoproteasomes and may support higher-affinity binding of proteasome inhibitors like PR-957. Furthermore, the 20S proteasome purified in the absence of ATP from dormant spores shows additional peptide-like densities near the proteolytic active sites, which the authors interpret as potentially inhibitory peptides that either represent indigestible fragments from an abundant

substrate or peptides highly expressed in the spore. These peptide densities are not observed in the 26S proteasomes purified in the presence of ATP from germinated spores.

The structure of the 26S proteasome complex confirmed the absence Rpn12, Rpn13, and Rpn15 (Sem1), as previously proposed based on genome analyses, and it identified or confirmed several subunit truncations, including those of Rpn2, Rpn3, and Rpt5, further supporting reductive evolution as a general principle for microsporidian macromolecular complexes.

Significant sequence deviations compared to eukaryotic orthologs had previously rendered some of the subunit annotations challenging, and this structural study helped distinguishing and identifying particular proteins, especially the alpha subunits 1, 4 and 5 of the 20S complex. Using these structurally validated *V. necatrix* proteins as queries for data-base searches of microsporidian proteasome subunits, the authors generated a phylogenetic tree that is consistent with trees previous derived from rRNA or whole-proteome analyses.

Although the presented studies provide a first glimpse into the structure of the microsporidian proteasome, they are to a large extent confirmatory of previous genome or proteome analyses and otherwise highly speculative regarding potential functional consequences of the observed differences compared to other eukaryotic proteasomes, without providing sufficient experimental evidence. In the absence of more detailed, higher resolution structural and/or functional data, there are no major new revelations that go significantly beyond of what was previously known or postulated and would justify publication in *Nature Communications*.

We thank the reviewer for their careful examination of our manuscript, and have adapted the text and figures to address the concerns on the overly speculative nature of our work. Further, we have now identified the peptide bound within the proteolytic cavity as an ortholog to the proteasome inhibitor PI31, and have significantly altered the focus of our manuscript in accordance with this impactful finding. Although the inhibitory function of PI31 is well described (PMID: 35927584; PMID: 10471803), previous studies have been unable to identify the biological conditions that favor PI31-proteasome interactions, and have therefore been unable to identify a precise biological and structural function for PI31. Our data suggest that, in microsporidia, the PI31-like protein (PI31L) is highly enriched in dormant spores and symmetrically binds within the catalytic cavities of purified 20S proteasomes. The well-resolved density and high coverage of PI31L in mass spectrometry data on *V. necatrix* spore proteasomes indicate that PI31L serves a hibernative role in microsporidia.

Interestingly, while our manuscript was under review, a yeast PI31-20S complex structure was published that demonstrated that PI31 simultaneously inhibits all three active sites via unique folds at each active site (PMID: 35927584). While this study was highly informative, the group utilized an $\alpha 3$ knockout strain of yeast to procure aberrant proteasomes enriched in PI31, resulting in structural insights that are potentially not physiologically relevant. In this case, our structural insights are both confirmatory and novel, as the microsporidian ortholog of PI31 adopts similar folds to the yeast ortholog at $\beta 2$ and $\beta 5$ active sites, but a completely novel fold at the $\beta 1$ site (**Figure 2**, described in Lines 165-196). The microsporidian ortholog is also significantly shorter and appears to follow an alternative route through the 20S core (**Figure 2**). We believe that these discoveries bolster the importance of our study, and hope that these novel findings allay concerns over other confirmatory aspects of the study.

Major points:

1) Based on their proteasome purifications from dormant spores in the absence of ATP and from germinated spores in the presence of ATP, the authors propose that only the germinated sporoplasm contains assembled 26S proteasomes, which lack the extra peptide-like density within the 20S proteolytic chamber. However, in the absence of appropriate controls it is unclear whether the purified complexes indeed reflect the scenario present in dormant versus germinated spores. Based on the provided data, it cannot be ruled out that some of the proteasomes were assembled as 26S or 30S complexes within the dormant spore, despite the absence of ATP, but dissociated during the purification process. Conversely, the 12 % assembled 26S or 30S complexes isolated from germinated sporoplasm may have assembled during the purification in the presence of ATP. The authors even acknowledge “that either the germination or the presence of ATP in the buffer facilitated the association of the RP and CP”, but the latter possibility was neither tested nor ruled out. Importantly, there is no experimental evidence at all for the authors’ claim that their data indicate a rapid reactivation of proteasomes after host infection.

As controls to assess the presence of 26S or 30S proteasomes in dormant and germinated spores and whether the observed peptide fragments in the 20S chamber are indeed expelled upon RP binding, the authors should purify proteasomes from dormant spores in the presence of ATP and from germinated spores in the absence of ATP. Importantly, independent of the purification conditions, the isolation by size-exclusion chromatography will not be suited for drawing any conclusions about the speed of proteasome reactivation after host infection, as claimed by the authors.

The reviewer is correct to note that a conclusion on the rate of reactivation of proteasomes is unsubstantiated, and we have removed the hypothesis that proteasomes are rapidly reactivated after host infection. Importantly, the successful identification of the peptide densities allows us to address issues arising from our variable biochemical preparations. Previous work demonstrates that PI31 selectively inhibits 20S proteasomes (PMID: 25332237) and inhibits the ATP-dependent formation of 26S proteasomes (PMID: 24770418), suggesting a combination of factors is required for the reactivation of proteasomes after hibernation. It also indicates that PI31 does not inhibit 26S proteasomes. With these findings in mind, we have made significant alterations to the text in both the introduction and discussion to clarify that a combination of factors is likely important. We have also rearranged the results section to emphasize that the goal of the sporoplasm/ATP purification was to isolate 26S proteasomes and 20S proteasomes lacking the additional active site densities (PI31L), and not to comment on the rate or mechanism of PI31L removal.

2) The authors propose that the additional densities observed in the catalytic chamber of 20S CP from dormant spores represent indigestible and potentially inhibitory peptide fragments that occlude proteolytic active sites by interacting with several of the substrate-binding cavities and avoiding direct contact with the S1 and S2 sites, with the peptide backbone tracing 4 - 5 Angstrom away from the catalytic threonines. However, these claims are not well supported by the presented figures, and the authors should show in Fig. 2 the pockets and their occupancy by peptide side chains (or lack thereof) in greater detail, not just some blobs. The claimed 2.8 Angstrom resolution should certainly allow a more detailed depiction. Can it be ruled out that higher flexibility or non-identical sequences of the bound peptides cause the lack of density in some areas? To support their hypothesis that these peptides are present only in dormant spore proteasomes, the authors should assess whether 20S proteasomes purified from germinated sporoplasm in the absence of ATP show this extra density.

We have now identified the bound peptidic density as the microsporidian ortholog the PI31, using a combination of density-informed motif searches, structural homology searches, and mass spectrometry data (See also responses to Reviewer #1 comments (1) through (6)). This interaction is now thoroughly described (Lines 158-196) and depicted (**Figure 2** and **Supplementary Fig. 3**) within the results section. As you suggest, the linkers that connect the various active site binding regions are most likely not visible in the density due to the high flexibility of these regions. In some cases, low resolution bridges can be seen between different segments, supporting the hypothesis that sections of PI31L connecting the well resolved extended segments are flexible rather than cleaved.

3) It is proposed that microsporidian 20S proteasomes use a different gate access mechanism. However, there appear to be much more similarities than differences compared for instance to the yeast *S. cerevisiae* proteasome. Although the microsporidian gate may use a short alpha-helical segment of alpha-5's N-terminus as a plug, while the N-terminus of yeast's alpha-5 is more unstructured, both gates primarily rely on the alpha-2 N-terminus for gate closure. Instead of (or in addition to) showing a superimposition of the *S. cerevisiae* open and *V. necatrix* closed gates in Fig. 4c, the authors should show a superimposition of both closed states, which are expected to look very similar. The authors' claim that the N-termini of alpha-1, 6, and 7 in the closed-state of microsporidian 20S resemble those in the open state of yeast 20S is not supported by the presented figures, which depict just low-resolution sausages for the subunit backbones. The authors also discuss Tyr7 and Pro15 of alpha-6, but none of these details are shown, neither in the main figures nor in the supplement.

We thank the reviewer for their thoughtful analysis of our microsporidian 20S gate structure. We believe that the N-termini of the α subunits deviate significantly from their yeast counterparts. In yeast, the $\alpha 3$ subunit plays an important role in gate closure, and deletions of the first nine amino acids are associated with 'permeable' proteasomes that display elevated levels of background proteolysis. Consistently, the truncation of the first seven residues in the *V. necatrix* $\alpha 3$ subunit results in a less stringently closed gate. These findings may provide an explanation for the biological role of PI31L in microsporidia. It is possible that PI31L is needed to inhibit background digestion in the less stringently closed 20S structures. We have altered the text to highlight the important findings surrounding the $\alpha 3$ subunit and the more opened 20S pore structure (Lines 250-254). Additionally, the YDR motifs and conserved proline residues play a vital role in stabilizing the open-gate conformation for 26S proteasomes in typical eukaryotic proteasomes. We believe it is important to note that these residues are largely absent in microsporidia, which suggests that their opened-gate conformation diverges from that of typical eukaryotes. Unfortunately, we do not have a sufficient number of 26S particles to capture an open-gate conformation, and have therefore removed comments concerning how the structure of the α subunits in the closed-gate conformation may mirror their open-gate conformations. We have also updated **Figure 4** to more clearly visualize the points described in the text, and added yeast - *V. necatrix* sequence alignments to **Supplementary Data 2** to facilitate comparisons between orthologous proteins.

4) The structure of the microsporidian 26S proteasome confirms the absence of Rpn12 and Rpn13, which was previously proposed based on genome analyses, and the authors also conclude that no ortholog of Rpn15/Sem1 is present. However, can the presence of a short Sem1-like peptide indeed be ruled out at an approximate resolution of > 15 Angstrom in this region of the proteasome lid? The very superficial depictions in Fig. 5 do not allow any assessment of this.

We appreciate the reviewer’s insight and have added a comment to indicate that Sem1/Rpn15 is small, comparatively unstructured, and in a poorly-resolved region of our map, resulting in insufficient evidence to claim it is missing in microsporidia. We have also altered statements in the abstract, introduction, and conclusion to be consistent with this uncertainty (Lines 28, 276-279, 357).

The authors also state that microsporidian Rpt5 lacks the N-loop from its N-terminal domain. Again, no details are shown, and the numbering (residues 107 – 144 in S.c. Rpt5) is incorrect. A structured N-loop has so far only been described for human Rpt5, where it comprises residues 99-119, which are equivalent to residues 88 – 114 in S.c. Rpt5. (And there is a typo in the PDB citation for the S.c. structure in figure 5, which needs to be 6FVU, not 6VFU.)

We greatly appreciate the reviewer’s attention to detail in consideration of our figures. We have corrected the typo in the PDB code. The highlighted structure in Figure 5c showed a large segment of Rpt1 that is missing in microsporidia, and happens to be very visible in the orientation depicted in Figure 5c. It had been erroneously annotated as Rpt5, and this annotation has now been removed. As you have noted, the Rpt5 N-loop described in humans would correlate closely to residues 88-114 in yeast. The absence of these residues is verified via sequence alignment of yeast and *V. necatrix* Rpt5, as below. We believe that making sequences available, as we have done in **Supplementary Data 2**, is sufficient detail to note the absence of this protein segment. We have added alignments to **Supplementary Data 2** to facilitate comparisons of yeast - *V. necatrix* orthologs. We have also clarified in the text that the stabilizing function of the N-loop of Rpt5 has been described in humans, and that a comparison of the two protein sequences is what revealed the absence of the N-loop in Rpt5 (Lines 297-301).

Sc_Rpt5	MATLEELDAQTLPGDDELQEIILNLSQTELQTRAKLLDNEIRIFRSELQRLSHENNVMLE	60
Vn_Rpt5	-----MSIEELKKYDADVENRSLEIEIVERTKLINSETRILQSRVNNIKHDTATKSA	51
	:: .: * : * : * : * : * : * : * : * : * : * : * : * : .	
Sc_Rpt5	KIKDNKEKIKNNRQLPYLVANVVEVMDMNEIEDKENSESTTQGGNVNLDNTAVGKAAVVK	120
Vn_Rpt5	QVAENMEKIRINKQLPYLVGNVVEVFDEG-----NGVVN	85
	: : * * * : * : * * * * : * * * * * : * * * : * * : * * :	
Sc_Rpt5	TSSRQTVFLPMVGLVDPDKLKPNDLVGVNKSYSYLILDTPSEFDSRVKAMEVDEKPTETY	180
Vn_Rpt5	TSTRITSYLPLMLGLVNEELKPGELVALHKETNIIFEKLPNNYDTKVKAMELDTKPDETY	145
	** : * * : * * : * * : : * * * : * * : * * : * * : * * : * * : * * * * *	
Sc_Rpt5	SDVGGLDKQIEELVEAIVLPMKRAKFKDMGIRAPKALMYGPPGTGKTLARACAAQTN	240
Vn_Rpt5	EDIGGLERQIEELNEAIVLPLTHPERFARLKIKPPKGLVLMYGPPGTGKTLMARACASKTN	205
	. * : * * : * * * * * * * * * * * : : * * : * * : * * * * * * * * * * : * * :	
Sc_Rpt5	ATFLKLAAPQLVQMYIGEGAKLVRDAFALAKEKAPTIIFIDEIDAIGTKRFDSEKSGDRE	300
Vn_Rpt5	ATFLKLAGPQLVQMYIGDGARLVRDAFALAKEKAPTIIFIDEIDAIGTKRVASDKTGDRE	265
	* * * * * . * . * : * * * * *	
Sc_Rpt5	VQRTMLELLNQLDGFSSDDRKVLAAATNRVDVLDPALLRSGRLDRKIEFPLPSEDSRAQI	360
Vn_Rpt5	VQRTMLELLNQLDGFTSHDNVKIIAATNRVDILDPALLRSGRLDRKIEFPLPNSSEGRKRI	325
	* . . : * * : *	
Sc_Rpt5	LQIHSRKMTTDDDINWQELARSTDEFNGAQLKAVTVEAGMIALRNGQSSVKHEDFVEGIS	420
Vn_Rpt5	LQIHSRKMNIEENVNFDLSRSTEGFNGAQCQKAVCVEAGMAALRKDKNQISQNDPMDGIL	385
	* * * * * * . : : * : * : * * : * * * * * * * * * * * * * * * * * * : * * : * * :	
Sc_Rpt5	EVQARKSKSVSFYA	434
Vn_Rpt5	EVLARKKSKLVYFT	399
	** * * * . . . : : :	

The authors propose that the absence of Rpn13 and Rpt5’s N-loop would have functional consequences for ubiquitin binding and cleavage, yet this is pure speculation and there is no supporting experimental evidence. The functional importance of Rpt5’s N-loop has never been

investigated for any eukaryotic proteasome, and mutations of Rpn13 in yeast *S. cerevisiae* show no significant phenotype on their own.

Although we did not perform any experiments describing the impact of Rpn13/N-loop deletion on the ubiquitin binding potential in microsporidia, our conclusions are based on previous findings. Previous data in yeast have demonstrated that mutations of Rpn13 lead to a two-fold reduction in ubiquitin affinity for proteasomes (PMID: 21095592), and that Rpn13 contributes to proteasome stability *in vitro* (PMID: 18497817). While the reviewer is correct that yeast Rpn13 knockout strains show no significant phenotype under ideal culturing conditions, we have not commented on a phenotypic outcome in microsporidia. Instead, we speculate that the removal of Rpn13 reduces the ubiquitin binding potential of microsporidian proteasomes (Line 292-297), in alignment with previous *in vitro* experimentation in yeast. We also make no claims on the functional implications of the N-loop deletions. We have changed the text to stress that deletions of Rpn13 in yeast are nonlethal, and emphasized that they are instead associated specifically with a change in proteasome affinity to ubiquitin (Lines 292-297). This finding is consistent with the take-home message of this section, which is that microsporidia have removed segments associated with important, but nonessential, functions.

The authors state that the absence of Rpn13 also results in the deletion of its binding partner UCH37, a deubiquitinase that serves a proofreading function for inadvertently ubiquitylated proteins at the proteasome. It is discussed that this is consistent with the hypothesis that editing and high-fidelity tasks are common targets for macromolecular streamlining in microsporidia. However, UCH37 is also lacking in *S. cerevisiae*, which seems perfectly fine without it.

We thank the reviewer for informing us about this important distinction between yeast and human proteasomes. We have removed the discussion of UCH37 and what its absence might mean for microsporidia.

5) Based on a comparison between six known conformational states of the yeast *S.c.* proteasome and quaternary subunit arrangements in the *V.n.* proteasome, the authors conclude that their structure is most similar to the s2 state. However, considering the great similarities between all conformations except for s1, this assessment appears not very reliable at a resolution of > 10 Angstrom for the ATPase ring and > 15 Angstrom for the lid, and it is certainly not convincing in the absence of any detailed figure representation. Given that the microsporidian proteasome most likely samples different conformational states similar to other eukaryotic proteasomes, it seems unfounded for the authors to postulate that the *V.n.* proteasome adopts a primed, but inactive conformation whose incompletely closed 20S gate “may allow for a more passive degradation of substrates, allowing microsporidia to conserve vital ATP.”

We have adjusted the text to emphasize the conformational diversity that is likely present in our sample (Lines 308-314), and removed speculation as to the functional significance of the “partially opened” CP gate in the 26S proteasome.

6) Regarding the numerous missing loops and truncated termini for several microsporidian proteasome subunits, has this been confirmed by sequencing in all cases or is this primarily based on structural information, where higher flexibility and lower local resolution may obscure more extended or unstructured segments? There are no legends or descriptions provided for

supplementary data 1 and 2, and it is unclear whether the sequences for the proteasome subunits listed in supplementary data 1 originate from mass spectrometry or genome sequencing.

The missing loops and truncated termini have been confirmed by whole genome sequencing of *V. necatrix* (manuscript in preparation). The legends for **Supplementary Data 1 and 2** have now been added to both Excel files and the first page of the Supplementary Data file.

7) The introduction mentions Rpn4 as a structural protein of the proteasome, which is incorrect.

We apologize for this error and have removed the comment that Rpn4 is a structural protein in the proteasome.

Overall, this manuscript contains a lot of speculations and draws numerous conclusions with respect to microsporidian proteasome assembly and function that are not supported by experimental evidence. Besides providing the first low-resolution structures and a confirmation of a reductive evolution for microsporidian

We have removed several of the highlighted claims in this revised version of our manuscript. More importantly, the focus of our manuscript lies now on the exciting new finding of the structural description of the minimal microsporidian PI31L in complex with the proteasome for which we have solid experimental data at 2.8 Å.

REVIEWERS' COMMENTS

Reviewer #1 (Remarks to the Author):

The authors have substantially revised the manuscript, very notably, with the addition of convincing identification of a novel PI31-like peptide and the newly added atomic model of this peptide. Together, the revised manuscript offers significant insights into both the function and evolution of PI31 and/or PI31-like protein and how they regulate the proteasome function in a tissue and species-dependent manner. I found the high-resolution structural analysis from the endogenous source of PI31-like protein from an important pathogen to be appealing and provide considerable insights into microsporidian proteasome. The revised Figure 2 is excellent. These novel, important results are well in line with the other study of PI31/Fub1-bound yeast 20S proteasome and provide interesting high-quality data beyond the scope of the existing studies on human and yeast proteasome, thus warranting its publication in Nature Communications. I therefore strongly recommend its publication in the present form.

Reviewer #3 (Remarks to the Author):

In their rebuttal and manuscript revision, Jespersen et al. have thoroughly addressed previous reviewers' criticism and suggestions. Importantly, identifying the additional density in 20S proteasomes from dormant spores as a V.n. ortholog of the inhibitor PI31 significantly boosted the impact and relevance of this study. Not only provides the modeling of PI31 bound to the wild-type V.n. 20S proteasome additional insights into the mechanism of proteolytic inhibition, these findings also reveal important details about the regulation of proteasome activity in dormant versus germinated spores, and represent a starting point for potential future development of inhibitors specific for microsporidian proteasomes.

This new, much stronger emphasis, together with the numerous edits, corrections, and improvements in response to the previous reviews, make this manuscript well suited for publication in Nature Communications.

Response to reviewers

We want to thank all reviewers for the critical reading of our manuscript, the constructive feedback, and the positive assessment of our work.

REVIEWERS' COMMENTS

Reviewer #1 (Remarks to the Author):

The authors have substantially revised the manuscript, very notably, with the addition of convincing identification of a novel PI31-like peptide and the newly added atomic model of this peptide. Together, the revised manuscript offers significant insights into both the function and evolution of PI31 and/or PI31-like protein and how they regulate the proteasome function in a tissue and species-dependent manner. I found the high-resolution structural analysis from the endogenous source of PI31-like protein from an important pathogen to be appealing and provide considerable insights into microsporidian proteasome. The revised Figure 2 is excellent. These novel, important results are well in line with the other study of PI31/Fub1-bound yeast 20S proteasome and provide interesting high-quality data beyond the scope of the existing studies on human and yeast proteasome, thus warranting its publication in Nature Communications. I therefore strongly recommend its publication in the present form.

Reviewer #3 (Remarks to the Author):

In their rebuttal and manuscript revision, Jespersen et al. have thoroughly addressed previous reviewers' criticism and suggestions. Importantly, identifying the additional density in 20S proteasomes from dormant spores as a *V.n.* ortholog of the inhibitor PI31 significantly boosted the impact and relevance of this study. Not only provides the modeling of PI31 bound to the wild-type *V.n.* 20S proteasome additional insights into the mechanism of proteolytic inhibition, these findings also reveal important details about the regulation of proteasome activity in dormant versus germinated spores, and represent a starting point for potential future development of inhibitors specific for microsporidian proteasomes. This new, much stronger emphasis, together with the numerous edits, corrections, and improvements in response to the previous reviews, make this manuscript well suited for publication in Nature Communications.